# NON-NEGATIVE CONTRASTIVE LEARNING

**Yifei Wang**[1]* **Qi Zhang**[2]* **Yaoyu Guo**[1] **Yisen Wang**[2, 3]†

[1] School of Mathematical Sciences, Peking University
[2] National Key Lab of General Artificial Intelligence,
School of Intelligence Science and Technology, Peking University
[3] Institute for Artificial Intelligence, Peking University

## ABSTRACT

Deep representations have shown promising performance when transferred to downstream tasks in a black-box manner. Yet, their inherent lack of interpretability remains a significant challenge, as these features are often opaque to human understanding. In this paper, we propose Non-negative Contrastive Learning (NCL), a renaissance of Non-negative Matrix Factorization (NMF) aimed at deriving interpretable features. The power of NCL lies in its enforcement of non-negativity constraints on features, reminiscent of NMF's capability to extract features that align closely with sample clusters. NCL not only aligns mathematically well with an NMF objective but also preserves NMF's interpretability attributes, resulting in a more sparse and disentangled representation compared to standard contrastive learning (CL). Theoretically, we establish guarantees on the identifiability and downstream generalization of NCL. Empirically, we show that these advantages enable NCL to outperform CL significantly on feature disentanglement, feature selection, as well as downstream classification tasks. At last, we show that NCL can be easily extended to other learning scenarios and benefit supervised learning as well. Code is available at `https://github.com/PKU-ML/non_neg`.



| (a) CL | (b) NCL | (c) CL | (d) NCL |

Figure 1: Feature visualization on semantic consistency (a-b) and sparsity (c-d) on CIFAR-10. The first two demonstrate top-activated samples along each feature dimension, where those of CL (a) often have distinct semantics along each dimension (column) (*e.g.,* dears and airplanes), while those of NCL (b) have much better semantic consistency, indicating better feature disentanglement. Comparing (c) and (d), it is easy to see that NCL features enjoy much better sparsity than CL features with only a few activated dimensions ($< 10\%$) per sample.

## 1 INTRODUCTION

It is widely believed that the success of deep learning lies in its ability to learn meaningful representations (Bengio et al., 2013). In recent years, contrastive learning (CL) has further initiated a huge interest in self-supervised learning (SSL) of representations and attained promising performance in various downstream tasks (Chen et al., 2020; Wang et al., 2021; Guo et al., 2023; Luo et al., 2023). Nevertheless, the learned representations still lack natural interpretability. As shown in Figure 1a, top activated examples along each feature dimension (column) belong to quite different classes, so we can hardly inspect the meaning of each feature. Although various interpretability tools have been

---

*Equal Contribution. Yifei Wang has graduated from Peking University, and is currently a postdoc at MIT.
†Corresponding Author: Yisen Wang (yisen.wang@pku.edu.cn).

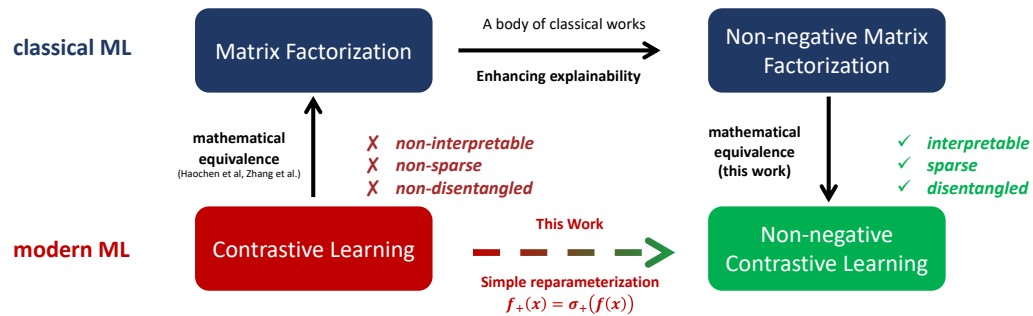

Figure 2: Relationship between different learning paradigms discussed in this work.

developed (Selvaraju et al., 2017), we are wondering whether there is a way to directly learn interpretable features while maintaining the simplicity, generality, and superiority advantages of deep representation learning.

In this paper, we develop Non-negative Contrastive Learning (NCL) as an interpretable alternative to canonical CL. Through a simple reparameterization, NCL can remarkably enhance the feature interpretability, sparsity, orthogonality, and disentanglement. As illustrated in Figure 1b, each feature derived from NCL corresponds to a cluster of samples with similar semantics, offering intuitive insights for human inspection. NCL draws inspiration from the venerable Non-negative Matrix Factorization (NMF) technique. Notably, by imposing non-negativity on features, NMF is known to derive components that are more naturally interpretable than its unrestricted counterpart, Matrix Factorization (MF) (Lee & Seung, 1999). Guided by this merit, we develop NCL as a non-negative counterpart for CL. NCL is indeed mathematically equivalent to the NMF objective, analogous to the equivalence between CL and MF discovered in prior work (HaoChen et al., 2021; Zhang et al., 2023a;b) (refer to Figure 2 for a visual summary). Built upon this connection, we theoretically justify these desirable properties of NCL in comparison to CL. In particular, we establish provable guarantees on the identifiability and downstream generalization of NCL, showing that NCL, in the ideal case, can even attain Bayes-optimal error. In practice, NCL can bring clear benefits to various tasks (feature selection, feature disentanglement, and downstream classification). At last, we show that NCL can be extended and yield better performance for supervised learning as well with the proposed Non-negative Cross Entropy (NCE) loss. We summarize our contributions as follows:

- We introduce **Non-negative Contrastive Learning (NCL)**, an evolved adaptation of NMF, to address the interpretability challenges posed by CL. Our findings demonstrate NCL's superiority in sparsity, orthogonality, and feature disentanglement.

- We provide a comprehensive theoretical analysis of NCL, which not only characterizes the optimal representations and justifies these desirable properties, but also establishes identifiability and generalization guarantees for NCL.

- We show how the advantages of NCL in feature interpretability can be applied to improving various downstream tasks. We also discuss how to extend NCL beyond SSL with a demonstration of its benefit in supervised learning.

## 2 BACKGROUND ON CONTRASTIVE LEARNING

The task of representation learning is to learn an encoder function $f : \mathbb{R}^d \to \mathbb{R}^k$ that extracts low-dimensional data representations $z \in \mathbb{R}^k$ (*a.k.a.* features) from inputs $x \in \mathbb{R}^d$. To perform contrastive learning, we first draw a pair of positive samples $(x, x_+)$ by randomly augmenting the same natural sample $\bar{x} \sim \mathcal{P}(\bar{x})$, and we denote the augmentation distribution as $\mathcal{A}(\cdot|\bar{x})$. We also draw negative samples $x^-$ as independently augmented samples, following the marginal distribution $\mathcal{P}(x) = \mathbb{E}_{\bar{x}} \mathcal{A}(x|\bar{x})$. The goal of contrastive learning is to align the features of positive samples while pushing negative samples apart. A well-known CL objective is the InfoNCE loss (Oord et al., 2018),

$$\mathcal{L}_{\mathrm{NCE}}(f) = -\mathbb{E}_{x,x_+,\{x_i^-\}_{i=1}^M} \log \frac{\exp(f(x)^\top f(x_+))}{\exp(f(x)^\top f(x_+)) + \sum_{i=1}^M \exp(f(x)^\top f(x_i^-))}, \tag{1}$$

where $\{x_i^-\}_{i=1}^M$ are $M$ negative samples independently drawn from $\mathcal{P}(x)$. HaoChen et al. (2021) propose the spectral contrastive loss (spectral loss) that is more amenable for the theoretical analysis:

$$\mathcal{L}_{\mathrm{sp}}(f) = -2\mathbb{E}_{x,x_+}f(x)^\top f(x_+) + \mathbb{E}_{x,x^-}(f(x)^\top f(x_i^-))^2. \tag{2}$$

Besides CL, it is also used for other types of SSL (Zhang et al., 2022; 2023a).

**Equivalence to Matrix Factorization.** Notably, HaoChen et al. (2021) show that the spectral contrastive loss (Eq. (2)) can be equivalently written as the following matrix factorization objective:

$$\mathcal{L}_{\mathrm{MF}} = \|\bar{A} - FF^\top\|^2, \text{ where } F_{x,:} = \sqrt{\mathcal{P}(x)}f(x). \tag{3}$$

Here, $\bar{A} = D^{-1/2}AD^{-1/2}$ denotes the normalized version of the co-occurrence matrix $A \in \mathbb{R}_+^{N \times N}$ of all augmented samples $x \in \mathcal{X}$ (assuming $|\mathcal{X}| = N$ for ease of exposition): $\forall x, x' \in \mathcal{X}, A_{x,x'} := \mathcal{P}(x,x') = \mathbb{E}_{\bar{x}}\mathcal{A}(x|\bar{x})\mathcal{A}(x'|\bar{x})$. More discussions on related work can be found in Appendix A.

## 2.1 LIMITATIONS IN REPRESENTATION SYMMETRY

According to the seminal work of Bengio et al. (2012), a good representation should extract explanatory factors that are sparse, disentangled, and with semantic meanings. However, upon close observation, CL-derived features remain predominantly non-explanatory and fall short of these criteria. For example, Figure 1a shows that different samples (like planes and dogs) may have similar activation along each feature dimension (a sign of non-disentanglement). Further, we also visualize the values of the learned features of SimCLR (Chen et al., 2020) in Figure 1c, where we can see that almost all CL features have non-zero values, indicating an absence of sparsity.

Remarkably, an important cause of CL's deficiency in these aspects is its *rotation symmetry*. Notice that CL objectives are invariant *w.r.t.* the rotation group $SO(k)$, which means that for an optimal solution $f^*(x)$, any rotation matrix $R$ also gives an optimal solution $Rf^*(x)$. Consequently, CL cannot guarantee feature disentanglement along a specific axis since a non-permutative rotation can easily corrupt them. The following theorem shows that any (unconstrained) CL objective depending only on pairwise distance will have rotation symmetry in its optimal solution, covering most (if not all) CL objectives, such as InfoNCE (Oord et al., 2018), hinge loss (Saunshi et al., 2019), NT-Xent (Chen et al., 2020), spectral contrastive loss (HaoChen et al., 2021)), *etc.*

**Theorem 1.** *As long as the unconstrained objective $\mathcal{L}$ only relies on pairwise Euclidean similarity (or distance), e.g., $f(x)^\top f(x')$, its solution $f^*(x)$ suffers from rotation symmetry.*

*Proof.* Observe that $\forall\, x, x' \in \mathcal{X}, (Rf^*(x))^\top Rf^*(x) = f^*(x)^\top R^\top Rf^*(x') = f^*(x)^\top f^*(x')$. $\square$

To attain feature disentanglement, we need to break this representation symmetry and learn representations aligned with the given axes.

## 3 NON-NEGATIVE CONTRASTIVE LEARNING

In classical machine learning, an effective way to make matrix factorization (MF) features interpretable is through non-negative matrix factorization (NMF), where a non-negative matrix $V \geq 0$ is factorized into two *non-negative* features $V \approx WH$ where $W, H \geq 0$ [1]. Inspired by the equivalence between MF and CL (Eq. (3)), we propose a modern variant for NMF (also with mathematical equivalence) and show that it also demonstrates good feature interpretability as NMF.

In the equivalent MF objective of CL, a critical observation is that the normalized co-occurrence matrix $\bar{A} \in \mathbb{R}_+^{N \times N}$ (Eq. (3)) is a **non-negative** symmetric matrix (all values in $\bar{A}$ are probability values). Therefore, we can apply symmetric NMF to $\bar{A}$ into non-negative features $F_+$:

$$\mathcal{L}_{\mathrm{NMF}}(F) = \|\bar{A} - F_+F_+^\top\|^2, \text{ where } F_+ \geq 0. \tag{4}$$

Unfornaturely, Eq. (4) is intractable with canonical NMF algorithms since the input space $\mathcal{X}$ is exponentially large ($N \to \infty$). To get rid of this obscure, we reformalize the NMF problem (Eq. (4)) equivalently as a tractable CL objective (Eq. (5)) that only requires sampling positive/negative samples from the joint/marginal distribution. The following theorem guarantees their equivalence.

---

[1] Throughout this paper, $\geq$ denotes element-wise comparison.

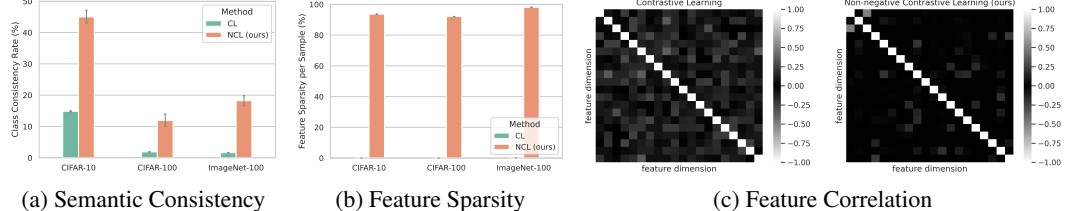

(a) Semantic Consistency  (b) Feature Sparsity  (c) Feature Correlation

Figure 3: Comaprisons between contrastive learning (CL) and non-negative contrastive learning (NCL): a) class consistency rate, measuring the proportion of activated samples that belong to their most frequent class along each feature dimension; b) feature sparsity, the average proportion of zero elements ($|x| < 1e^{-5}$) in the features of each test sample; c) dimensional correlation matrix $C$ of 20 random features: $\forall (i,j), C_{ij} = \mathbb{E}_x \tilde{f}_i(x)^\top \tilde{f}_j(x)$, where $\tilde{f}_i(x) = f_i(x)/\sqrt{\sum_x (f_i(x))^2}$.

**Theorem 2.** *The non-negative matrix factorization problem in Eq. (4) is equivalent to the following non-negative spectral contrastive loss when the row vector $(F_+)_{x,:} = \sqrt{\mathcal{P}(x)} f_+(x)^\top$,*

$$\mathcal{L}_{\text{NCL}} = -2\mathbb{E}_{x,x_+} f_+(x)^\top f_+(x_+) + \mathbb{E}_{x,x^-} \left( f_+(x)^\top f_+(x^-) \right)^2,$$

$$\text{such that } f_+(x) \geq 0, \forall x \in \mathcal{X}. \tag{5}$$

**Non-negative Reparameterization.** Comparing Eq. (5) to the original spectral contrastive loss in Eq. (2), we find that the objectives remain the same, while the only difference is that NCL enforces non-negativity on output features $f_+(\cdot)$. We can realize this constraint via a reparameterization of neural network outputs, *e.g.,* applying a non-negative transformation $\sigma_+(\cdot)$ (satisfying $\sigma_+(x) \geq 0, \forall x$) at the end of a standard neural network encoder $f$ used in CL,

$$f_+(x) = \sigma_+(f(x)). \tag{6}$$

Potential choices of $\sigma_+$ include $\text{ReLU}, \text{sigmoid}, \text{softplus}$, *etc.* We find that the simple $\text{ReLU}$ function, $\text{ReLU}(x) = \max(x,0)$, works well in practice. Its zero-out effect can induce good feature sparsity compared to other operators with strictly positive outputs. Although our discussions mainly adopt the spectral contrastive loss, this reparameterization trick also applies to other CL objectives (such as the InfoNCE loss in Eq. (1)) and transforms them to the corresponding non-negative versions. We regard these variants generally as Non-negative Contrastive Learning (NCL). We also briefly discuss how to extend it to other learning paradigms (supervised, multi-modal) in Section 6.

**Resurrecting the Dead Neurons.** We notice that although ReLU output features enjoy better sparisty, they may suffer from the dead neuron problem during training (Lu et al., 2019). In practice, some dimensions of NCL features (*e.g.,* 86/512) are inactive on all samples, sacrificing the given model capacity. To avoid this problem while ensuring non-negativity, inspired by Gumbel softmax (Jang et al., 2016), we adopt a simple (biased) reparameterization trick (in PyTorch style),

```
z = F.relu(z).detach() + F.gelu(z) - F.gelu(z).detach(),        (7)
```

where `z.detach()` eliminates the gradients of z. In this way, the forward output still equals to `F.relu(z)` while the gradient is calculated *w.r.t.* GELU (Hendrycks & Gimpel, 2016), which is close to ReLU but has gradients for negative inputs. This reparameterization trick works on CIFAR-10 and CIFAR-100 and improves the overall feature diversity, but is inferior to ReLU on ImageNet-100. More systematic study of this problem is left for future work.

### 3.1 BENEFITS OF NON-NEGATIVITY: CONSISTENCY, SPARSITY, AND ORTHOGONALITY

In canonical NMF, researchers found that non-negative constraints help discover interpretable parts of facial images (Lee & Seung, 1999). It also has applications in many scenarios for discovering meaningful features (Wang & Zhang, 2012), such as text mining, factor analysis, and speech denoising. In our scenario, the non-negativity inherent to NCL offers clear advantages in deriving interpretable representations. Unlike canonical CL, non-negativity ensures that different features will not cancel one another out when calculating feature similarity. As NCL drives dissimilar negative samples to have minimal similarity ($f_+(x)^\top f_+(x') \to 0$), each sample is forced to have zero activation on features it does not possess, rendering NCL features inherently sparse.

Furthermore, NCL offers a robust solution to the rotation symmetry challenge associated with standard CL, as outlined in Theorem 1. Any rotation applied to features, especially sparse ones, often disrupts their non-negative nature. To exemplify, consider the feature matrix $I = \begin{pmatrix} 1 & 0 \\ 0 & 1 \end{pmatrix}$. This matrix cannot undergo any non-permutative rotation while preserving its non-negativity. As a consequence, NCL features are enforced to be aligned with the given axes. In particular, with non-negative constraints, activated samples along each feature dimension have high similarity and thus have similar semantics.

To verify the analysis above, we conduct a pilot study on SimCLR and its NCL variant on benchmark datasets. First, we find that NCL features indeed have better semantic consistency than CL (Figure 3a), aligning with the examples in Figure 1. Second, around 90% dimensions of NCL features are zero *per sample*, while CL features do not have any sparsity, as we observed in Figure 3b. Third, Figure 3c shows the feature correlation matrices, and we can observe that NCL features are also more orthogonal than CL features. Overall, NCL enjoys much better semantic consistency, feature sparsity, and feature orthogonality than CL.

## 4 THEORETICAL PROPERTIES OF NON-NEGATIVE CONTRASTIVE LEARNING

In Section 3, we have proposed non-negative contrastive learning (NCL) and shown that a simple reparameterization can bring significantly better semantic consistency, sparsity, and orthogonality. In this section, we provide comprehensive theoretical analyses of non-negative contrastive learning in terms of its optimal representations, feature identifiability, and downstream generalization.

### 4.1 ASSUMPTIONS

As in Saunshi et al. (2019), we assume that in the pretraining data, samples belong to $m$ *latent classes* $\mathcal{C} = \{c_1, \dots, c_m\}$ with non-neglectable support $\mathcal{P}(c) > 0, \forall c \in \mathcal{C}$. We assume that positive samples generated by contrastive learning are drawn independently from the same latent class.

**Assumption 1** (Positive Generation). $\forall x, x' \in \mathcal{X}, \mathcal{P}(x, x') = \mathbb{E}_c \mathcal{P}(x|c)\mathcal{P}(x'|c)$.

**Remark.** Some previous works, *e.g.,* Wang et al. (2022), criticized this assumption to be non-realistic. But this is mainly because Saunshi et al. (2019) only consider the extreme case when the latent classes are the same as the observed labels, *i.e.,* $\mathcal{C} = \mathcal{Y}$; while in fact, positive samples generated by data augmentations are still dependent within the same observed class. In fact, when we define latent classes using more fine-grained categorization (*e.g.,* a specific type of car), this assumption is still plausible. In the extreme case, each original sample $\bar{x} \in \mathcal{X}$ can be seen as a latent class, which reduces to the general definition in HaoChen et al. (2021) with $\mathcal{P}(x, x') = \mathbb{E}_{\bar{x}} \mathcal{P}(x|\bar{x})\mathcal{P}(x'|\bar{x})$. Thus, our framework includes the two previous frameworks as special cases.

As a common property of natural data, we assume different latent classes only have small overlap.

**Assumption 2** (Latent Class Overlap). *The maximal class overlap probability is no more than $\varepsilon$, i.e.,* $\max_{i \neq j} \mathcal{P}(c_i, c_j) \leq \varepsilon$, *where* $\mathcal{P}(c_i, c_j) = \mathbb{E}_x \mathcal{P}(c_i|x)\mathcal{P}(c_j|x)$ *denotes the overlap probability between $c_i$ and $c_j$. If each sample only belongs to one latent class, we have $\varepsilon = 0$.*

The latent class information of an example $x$ can be denoted as an $m$-dimensional vector $\psi(x) = [\mathcal{P}(c_1|x), \dots, \mathcal{P}(c_k|x)]$. This vector captures the subclass-level semantics and can be seen as an ideal representation of $x$. Next, we show that non-negative contrastive learning can recover this ground-truth feature from observed samples, up to trivial dimensional scaling and permutation.

### 4.2 OPTIMAL REPRESENTATIONS

The following theorem states that when the number of feature dimensions $k$ is as large as the number of latent classes $m$, the following solution $\phi(x)$ is an optimal solution to the NCL objective:

$$\phi(x) = \left[ \frac{1}{\sqrt{\mathcal{P}(\pi_1)}} \mathcal{P}(\pi_1|x), \dots, \frac{1}{\sqrt{\mathcal{P}(\pi_m)}} \mathcal{P}(\pi_m|x) \right] \in \mathbb{R}_+^m, \forall \, x \in \mathcal{X}, \tag{8}$$

where $[\pi_1, \ldots, \pi_m]$ is a random permutation of latent classes $[c_1, \ldots, c_m]$.[2] Nicely, $\phi(x)$ admits natural interpretability since it contains the posterior distribution along each dimension.

**Theorem 3.** *Under the latent class assumption (Assumption 1) and choosing $k = m$, $\phi(\cdot)$ is a minimizer of the NCL objective Eq. (5), i.e., $\phi \in \arg\min \mathcal{L}_{\mathrm{NCL}}$.*

Compared to CL with non-explanatory eigenvectors of $\bar{A}$ as the closed-form solution (HaoChen et al., 2021), NCL's optimal features $\phi(x)$ are advantages in the following aspects. **1) Semantic Consistency.** For each $c \in \mathcal{C}$, the activated samples with $\phi_c(x) > 0$ indicates that $\mathcal{P}(c|x) > 0$, meaning that $x$ indeed belongs to $c$ to some extent. Thus, activated samples along each dimension will have similar semantics. Instead, CL's eigenvector features containing both positive and negative values do not have a clear semantic interpretation. **2) Sparsity.** As each sample belongs to one or a few subclasses (Assumption 2), only one or a few elements in $\phi(x)$ are positive with $\mathcal{P}(c|x) > 0$. Thus, $\phi(x)$ is sparse. In comparison, eigenvectors used by CL are usually dense vectors. **3) Orthogonality.** Since feature dimensions in $\phi(x)$ indicate class assignments, that different feature dimensions have very low correlation when the latent class overlap $\varepsilon$ is small (Assumption 2):

$$\forall\, i \neq j, \mathbb{E}_x \phi_i(x)\phi_j(x) = \frac{1}{\sqrt{\mathcal{P}(\pi_i)\mathcal{P}(\pi_j)}} \mathbb{E}_x \mathcal{P}(\pi_i|x)\mathcal{P}(\pi_j|x) \leq \frac{\varepsilon}{\min_c \mathcal{P}(c)}. \qquad (9)$$

In the ideal case when evey sample belongs to a single latent class, we can show that $\phi(x)$ perfectly recovers the one-hot ground-truth factors with high sparsity and perfect orthogonality.

**Theorem 4** (Optimal representations under one-hot latent labels). *If each sample $x$ only belongs to only one latent class $c = \mu(x)$, we have $\phi(x) = \mathbf{1}_{\mu(x)}$, where $\mu(x)$ indicates $x$'s belonging class. Meanwhile, we have $\|\phi(x)\|_0 = 1$ (highly sparse) and $\mathbb{E}_x \phi(x)\phi(x)^\top = I$ (perfectly orthogonal).*

### 4.3 FEATURE IDENTIFIABILITY

Despite the advantages of $\phi(x)$ discussed above, it remains unclear whether it is the *unique* solution to NCL. As discussed in Section 2.1, rotation symmetry will break the desirable properties of $\phi(x)$. In a general context, the uniqueness of NMF is an important research topic in itself (Laurberg et al., 2008; Huang et al., 2014). It is also widely studied in signal analysis and representation learning, known as (feature) identifiability (Fu et al., 2018; Zhang et al., 2023b), concerning whether NMF can recover ground-truth latent factors (up to trivial transformation), as defined below.

**Definition 1** (Feature Identifiability). *We say that an algorithm produces identifiable (or unique) features if any two solutions $f, g$ are equivalent up to permutation and dimensional scaling: there exists a permutation matrix $P$ and a diagonal matrix $D$ such that $f(x) = DPg(x), \forall\, x \in \mathcal{X}$.*

For NCL, we find that a mild condition on unique samples (Assumption 3) can guarantee NCL's identifiability and disentanglement. This assumption is plausible in practice since it only requires one unique sample in each latent class. Intuitively, the one-hot encodings of unique samples prevent any non-permutative rotation under the non-negative constraint. Therefore, NCL's non-negativity indeed helps break the rotation symmetry of CL features.

**Assumption 3** (Existence of unique samples). *For every latent class $c \in \mathcal{C}$, there exists one sample $x \in \mathcal{X}$ such that it belongs uniquely to $c$, i.e., $P(c|x) = 1$.*

**Theorem 5.** *Under Assumptions 1 & 3, the solution $\phi(x)$ (Eq. (8)) is the unique solution to the NCL objective Eq. (5). As a result, NCL features are identifiable and disentangled.*

### 4.4 DOWNSTREAM GENERALIZATION

In real-world applications, pretrained features are often transferred to facilitate various downstream tasks. Following the previous convention, we consider the linear classification task, where a linear head $g(z) = W^\top z$ is applied to pretrained features $z = \phi(x)$ (Eq. (8)). The linear head is learned

---

[2]For the ease of exposure, we consider $k = m$. If $k > m$, we can fill the rest of dimensions with all zeros, while if $k < m$, we can always tune $k$ to attain a lower loss. In practice, some feature dimensions are never activated, so we believe that generally we have $k > m$ and omit inactive dimensions for simplicity. Besides, although the discrete latent class is adopted here, the discovered optimal solutions also apply to continuous latent variables, as long as the position generation still follows Assumption 1.

Table 1: Comparison of CL and NCL (ours) on ImageNet-100 in terms of three feature selection tasks: linear probing (measured by classification accuracy), image retrieval (measured by mean Average Precision @ 10 (mAP@10), and transfer learning (measured by classification accuracy).

| Selection | Linear Probing | | Image Retrieval | | Transfer Learning | |
|---|---|---|---|---|---|---|
| | CL | NCL | CL | NCL | CL | NCL |
| All (2048) | $66.8 \pm 0.2$ | $\mathbf{68.9 \pm 0.1}$ | $10.9 \pm 0.2$ | $\mathbf{14.2 \pm 0.2}$ | $17.2 \pm 0.1$ | $\mathbf{19.9 \pm 0.1}$ |
| Random (512) | $66.2 \pm 0.1$ (-0.6) | $64.3 \pm 0.2$ (-5.6) | $10 \pm 0.1$ (-0.9) | $8.2 \pm 0.1$ (-6.0) | $16.6 \pm 0.2$ (-0.6) | $16.7 \pm 0.1$ (-3.2) |
| EA (512, w/o ReLU) | $66.3 \pm 0.2$ (-0.5) | $66.7 \pm 0.1$ (-2.2) | $9.9 \pm 0.21$ (-0.9) | $11.1 \pm 0.2$ (-3.1) | $16.5 \pm 0.3$ (-0.7) | $17.7 \pm 0.2$ (-2.2) |
| **EA (512, w/ ReLU) (ours)** | $66.5 \pm 0.1$ (-0.3) | $\mathbf{68.9 \pm 0.3}$ (-0.0) | $10.2 \pm 0.2$ (-0.7) | $\mathbf{14.2 \pm 0.2}$ (-0.0) | $16.6 \pm 0.3$ (-0.6) | $\mathbf{19.8 \pm 0.1}$ (-0.1) |

on the labeled data $(x, y) \sim \mathcal{P}(x, y)$, where $y \in \mathcal{Y} = \{y_1, \ldots, y_C\}$ denotes an observed label. As discussed in Section 4.1, latent classes are supposed to correspond to more fine-grained clusters within each observed class. For example, an observed class "dog" contains many species of dogs as its latent classes. Thus, we assume that each observed label $y$ contains a set of latent classes $\mathcal{C}_y \subseteq \mathcal{C}$, and there is no overlap between different observed labels, *i.e.*, $\forall\, i \neq j, \mathcal{C}_{y_i} \bigcap \mathcal{C}_{y_j} = \emptyset$.

The following theorem characterizes the optimal linear classifier under this setting, and shows that the linear classifier alone can attain the Bayes optimal classifier, $\arg\max_y \mathcal{P}(y|x)$. In other words, ideally, a linear classifier is enough for the classification task. Nevertheless, in practice, since the learning process is not as perfect as we have assumed, fully fine-tuning usually brings further gains.

**Theorem 6.** *With the optimal NCL features $\phi(x)$, there exists a linear classifier $g^*(z) = W^{*\top} z$ such that its prediction $p(x) = \arg\max_y [g(\phi(x))]_y$ is **Bayes optimal**, specifically,*

$$W^* = [w_1^*, \ldots, w_C^*], \ where\ w_y^* = [\sqrt{\mathcal{P}(\pi_1)} 1_{\pi_1 \in \mathcal{C}_y}, \ldots, \sqrt{\mathcal{P}(\pi_m)} 1_{\pi_m \in \mathcal{C}_y}], \forall y \in [C]. \quad (10)$$

The optimal linear classifier $g^*(z)$ also enjoys good explanability, since only latent classes in $\mathcal{C}_y$ have non-zero weights in $w_y$. Therefore, we can directly inspect how each latent class $c$ belongs to a certain observed class $y$ by observing non-zero weights in $W^*$.

## 5 APPLICATIONS

In this section, leveraging the ability of non-negative contrastive learning (NCL) to extract explanatory features, we explore three application scenarios: feature selection, feature disentanglement, and downstream generalization. In the former two scenarios, NCL brings significant benefits over canonical CL. In the standard evaluation with downstream classification, NCL also obtains better performance than CL under both linear probing and fine-tuning settings. By default, we compare CL and NCL objectives using the SimCLR backbone on ImageNet-100 with 200 training epochs. For all experiments, we run 3 random trials and report their mean and standard deviation.

### 5.1 FEATURE SELECTION

As shown in Figure 3b, NCL features are only sparsely activated on a few feature dimensions, and upon our closer examination, many feature dimensions are only activated with a few examples. Leveraging this feature sparsity property, we can sort out feature importance with non-negative features and use this as a guidance for selecting a small subset of features while retaining most of the performance. This is particularly useful for large-scale image or text retrieval tasks by allowing a flexible tradeoff between performance and computation cost, a natural way to realize *shortening embeddings* (OpenAI, 2024) without additional training strategies as in Kusupati et al. (2022).

Here, we explore a simple strategy to rank non-negative features $f = [f_1, \ldots, f_k]$ according to their Expected Activation (EA) on the training data, $\mathrm{EA}(f_i) = \mathbb{E}_x \tilde{f}_i(x)$, where $\tilde{f}(x) = f(x)/\|f(x)\|$ is normalized. Intuitively, higher EA indicates common high-level features that represent class semantics, and lower EA corresponds to rare and noisy features that are less useful. In our experiments, we select the top 512 features among all 2048 features of projector outputs, and evaluate them with linear probing, image retrieval, and transfer learning on ImageNet-100 (details in Appendix G.4).

**Results.** Table 1 summarizes the results of feature selection experiments. First, when using all features, NCL achieves significant gains over CL on all three tasks. Meanwhile, when selecting 1/4 dimensions (512/2048), with EA criterion (w/ ReLU), NCL is almost "lossless", since it attains

Table 2: Feature disentanglement score (measured by SEPIN@$k$ (Do & Tran, 2020)) of CL and NCL on ImageNet-100, where $k$ denotes the top-$k$ dimensions. Values are scaled by $10^2$.

|  | SEPIN@1 | SEPIN@10 | SEPIN@100 | SEPIN@all |
|---|---|---|---|---|
| CL | $0.88 \pm 0.08$ | $0.79 \pm 0.02$ | $0.69 \pm 0.01$ | $0.47 \pm 0.01$ |
| **NCL** | $\mathbf{7.43 \pm 0.15}$ | $\mathbf{5.93 \pm 0.12}$ | $\mathbf{3.87 \pm 0.04}$ | $\mathbf{0.48 \pm 0.01}$ |

Table 3: Evaluation of learned representations on linear probing (LP) and finetuning (FT) tasks. (a) in-distribution evaluation on three benchmark datasets: CIFAR-100, CIFAR-10, and ImageNet-100. (b) linear probing accuracy of ImageNet-100 pretrained features on three OOD datasets.

(a) in-distribution evaluation

| Method | CIFAR-100 | | CIFAR-10 | | ImageNet-100 | |
|---|---|---|---|---|---|---|
|  | LP | FT | LP | FT | LP | FT |
| CL | $58.6 \pm 0.2$ | $72.6 \pm 0.1$ | $87.6 \pm 0.2$ | $92.3 \pm 0.1$ | $68.7 \pm 0.3$ | $77.3 \pm 0.5$ |
| NCL | $\mathbf{59.7 \pm 0.4}$ | $\mathbf{73.0 \pm 0.2}$ | $\mathbf{87.8 \pm 0.2}$ | $\mathbf{92.6 \pm 0.1}$ | $\mathbf{69.4 \pm 0.3}$ | $\mathbf{79.2 \pm 0.4}$ |

(b) out-of-distribution transferability

| Method | Stylized | Corruption | Sketch |
|---|---|---|---|
| CL | $19.6 \pm 0.4$ | $34.5 \pm 0.2$ | $27.1 \pm 0.1$ |
| NCL | $\mathbf{21.2 \pm 0.2}$ | $\mathbf{36.1 \pm 0.3}$ | $\mathbf{28.0 \pm 0.2}$ |

comparable performance to using all features on all tasks. In comparison, CL features always suffer from a larger drop when using fewer features. Thus, benefiting from the sparsity and disentanglement properties, NCL allows flexible feature selection that retains most of the original performance.

## 5.2 FEATURE DISENTANGLEMENT

According to Section 4.3, NCL features enjoy identifiability (under mild conditions), which is often regarded as a sufficient condition to achieve feature disentanglement (Khemakhem et al., 2020), while CL suffers from severe feature ambiguities (Theorem 1). To validate this property, we show that NCL can significantly improve the degree of feature disentanglement on real-world data.

**Evaluation Metric.** Many disentanglement metrics like MIG (Chen et al., 2018) are supervised and require ground-truth factors that are not available in practice, so current disentanglement evaluation is often limited to synthetic data (Zaidi et al., 2020). To validate the disentanglement on real-world data, we adopt an unsupervised disentanglement metric SEPIN@$k$ (Do & Tran, 2020). SEPIN@$k$ measures how each feature $f_i(x)$ is disentangled from others $f_{\neq i}(x)$ by computing their conditional mutual information with the top $k$ features, *i.e.,* SEPIN@$k = \frac{1}{k} \sum_{i=1}^{k} I(x, f_{r_i}(x) | f_{\neq r_i}(x))$, which are estimated with InfoNCE lower bound (Oord et al., 2018) in practice (see Appendix G.3).

**Results.** As shown in Table 2, NCL features show much better disentanglement than CL in all top-$k$ dimensions. The advantage is larger by considering the top features (*e.g.,* 0.69 *v.s.* 3.87 SEPIN@100), since learned features also contain noisy dimensions. This verifies our identifiability result that with non-negativity constraints, NCL indeed brings better feature disentanglement on real-world data.

## 5.3 DOWNSTREAM GENERALIZATION

Finally, we evaluate CL and NCL on the linear probing and fine-tuning tasks using all features on three benchmark datasets: CIFAR-10, CIFAR-100 (Krizhevsky et al., 2009), and ImageNet-100 (Deng et al., 2009). Apart from in-distribution evaluation, we also compare them on three out-of-distribution datasets, stylized ImageNet (Geirhos et al., 2018), ImageNet-Sketch (Wang et al., 2019), ImageNet-C (Hendrycks & Dietterich, 2019) (restricted to ImageNet-100 classes). More details can be found in Appendix G.5. Following the conventional setup, we adopt the encoder output for better downstream performance. See more results in Appendix F.4.

**Results.** As shown in Table 3, NCL shows superior generalization performance than original contrastive learning across multiple real-world datasets. On average, it improves linear probing accuracy by $0.6\%$ and improves fine-tuning accuracy by $0.9\%$. Moreover, we observe from Table 3b that NCL shows significant advantages in the out-of-distribution generalization downstream tasks ($+1.4\%$ on average). Considering that NCL only brings minimal change to CL by adding a simple reparameterization with neglectable cost, the improvements are quite favorable. It also aligns well

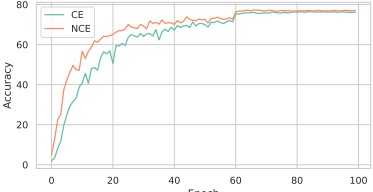

Figure 4: Training from scratch with CE and NCE (w/o projector) on ImageNet-100.

Table 4: Test accuracy (%) of CE and NCE losses for supervised learning on ImageNet-100.

| Loss | From Scratch | Finetune |
|---|---|---|
| CE | 76.1 | 78.6 |
| NCE | 78.6 | 80.2 |
| CE + MLP projector | 78.4 | 81.1 |
| NCE + MLP projector | **79.2** | **82.0** |

with the common belief that disentangled feature learning can yield representations that are more robust against domain shifts (Guillaume et al., 2019).

## 6 EXTENSION TO BROADER SCENARIOS

In the above discussions, we have developed NCL in the context of self-supervised learning, where contrastive learning originates from (Oord et al., 2018). Meanwhile, contrastive learning also finds wide applications in many other scenarios, such as, SupCon (Khosla et al., 2020; Cui et al., 2023) for supervised learning and CLIP (Radford et al., 2021a) for multi-modal learning. The generality of NCL enables easy extension. First, it is straightforward to generalize NCL to supervised contrastive learning since they adopt the same contrastive loss. Second, Zhang et al. (2023a) shows that multi-modal contrastive learning (MMCL) like CLIP is equivalent to *asymmetric* matrix factorization, $\|A - F_V F_L^\top\|^2$. Leveraging this connection, we can similarly derive multi-modal NCL as equivalent to asymmetric NMF (Appendix C). We leave a systematic study on broad domains to future work.

**Non-negative Cross Entropy (NCE).** Besides existing variants of contrastive learning, we also note that NCL can be extended to standard supervised learning with cross-entropy (CE) loss. Notice that the CE loss can be written as $\mathcal{L}_{\text{CE}}(f) = \mathbb{E}_{x,y} \log \frac{\exp(f(x)^\top w_y)}{\sum_{c=1}^C \exp(f(x)^\top w_c)}$, where $f(x)$ is the representation of $x$ and $W = [w_1, \ldots, w_C]$ contains the learned embeddings of each class in the last linear layer of $C$-way classification NNs. Thus, CE loss can be seen as a special case of InfoNCE when taking $(x, y)$ as positive pairs and $(f(x), w_y)$ as their representations (*i.e.,* a special MMCL loss (Appendix C)). Accordingly, we can derive the Non-negative Cross Entropy (NCE) loss by applying a non-negative transformation (*e.g.,* ReLU) to $(f(x), w_y)$ in CE and eliminate rotation symmetry.

**Setup.** As a preliminary study, we compare CE and NCE on ImageNet-100 for 1) finetuning a standard SimCLR-pretrained encoder and 2) training from scratch, with a ResNet-18 backbone (w/o projector). We also study adding an MLP projector (as in SimCLR) before the last linear layer, allowing more flexible feature transformations before linear classification (details in Appendix G.6).

**Results.** Figure 4 shows that non-negative training with NCE is much faster than conventional CE loss at the early stage (around 2x before the 40th epoch). After training converges, NCL also enjoys slightly better performance. As shown in Table 4, in terms of the final performance, NCE is helpful for training from scratch (2.5%). With the help of the MLP projector network, NCL also attains significant gains by outperforming original CE by 3.1% and CE-with-projector by 0.8%. For finetuning, with MLP projector, NCE can attain a significant gain by improving 3.4% over CE and 0.9 over CE-with projector. These results show that although simple, non-negative training can be quite helpful for supervised tasks as well.

## 7 CONCLUSION

Despite the promising performance of contrastive learning, its learned features still have limited interpretability. Inspired by the classical non-negative matrix factorization algorithm, we proposed non-negative contrastive learning (NCL) to impose non-negativity on contrastive features. With minimal modifications to canonical CL, NCL can maintain or slightly improve its performance on classical tasks, while significantly enriching the interpretability of output features. We also provided comprehensive theoretical analyses for NCL, showing that NCL enjoys not only good downstream performance but also feature identifiability. At last, we show that NCL can be extended to benefit other learning scenarios as well. Overall, we believe that NCL may serve as a better alternative to CL in pursuit of better representation learning methodologies.

ACKNOWLEDGEMENT

Yisen Wang was supported by National Key R&D Program of China (2022ZD0160300), National Natural Science Foundation of China (62376010, 92370129), and Beijing Nova Program (20230484344).

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

# A    RELATED WORK

The most closely related work is HaoChen et al. (2021), which firstly links contrastive learning back to matrix factorization. This link inspires us to leverage the classical non-negative matrix factorization technique, in turn, to improve modern contrastive learning. Previously, matrix factorization has played an important part in representation learning, for example, GloVe for word representation (Pennington et al., 2014), but was gradually replaced by deep models. Aside from our NMF-based approach, many works study disentangled representation with generative models like VAEs (Kingma & Welling, 2014; Chen et al., 2018). Locatello et al. (2019) point that disentanglement is impossible without additional assumptions (Kivva et al., 2022) or supervision (Khemakhem et al., 2020). Nevertheless, these methods often have limited representation ability compared to contrastive learning in practice. In this work, our goal is to develop a representation algorithm whose performance is as good as contrastive learning, while alleviating its interpretability limitations in real-world scenarios.

**Relationship to CL.** A deep connection between CL and NCL is that they are both equivalent to matrix factorization problems *w.r.t.* a non-negative cooccurrence matrix (Eq. (3) & Eq. (4)); while the difference in non-negative constraints leads to different solutions, and it enables good explainability of NCL features. Theoretically, CL admits eigenvectors as closed-form solutions (though not computationally tractable), while NMF (NCL) problems generally do not permit closed-form solutions. Yet, we show that under appropriate data assumptions regarding latent classes, we can still identify an optimal solution, and establish its uniqueness and generalization guarantees under certain conditions. Thus, for real-world data obeying our assumptions, NCL features are also guaranteed to deliver good performance, while offering significantly better explainability than canonical CL as discussed above.

**Relationship to Clustering.** In fact, Ding et al. (2005) also establish the equivalence between NMF, Kernel K-means and spectral clustering (SC)[3]. Leveraging this connection, NCL can also be regarded as an (implicit) deep clustering method. Previously, several works (HaoChen et al., 2021; Tan et al., 2023) claim the equivalence between canonical CL and SC. However, this connection is not rigorous, because they only consider the equivalence between CL and eigendecomposition (the first step of SC), while SC also has a K-means step that utilizes eigenvectors to produce cluster assignments as its final outputs (like NCL).[4]

**Relationship to MF and Deep NMF.** Matrix Factorization (MF) is a classical dimension reduction technique, with mathematical equivalence to many machine learning techniques, such as, PCA, LDA. However, it is pointed out that features extracted by MF are often hardly visually inspected. A seminal work by Lee & Seung (1999) points that by enforcing non-negativity constraints on MF, *i.e.,* non-negative matrix factorization (NMF), we can obtain interpretable part-based features. In their well-known face recognition experiment, each decomposed feature is only activated on a certain facial region, while PCA components have no clear meanings. Ding et al. (2005) further reveal the inherent clustering property of NMF, specifically, the equivalence between NMF and K-means clustering with additional orthogonal constraints. In this way, each NMF feature can be interpreted as the cluster assignment probability. Due to these advantages, NMF is widely applied to many scenarios, including but not limited to recommendation systems, computer vision, document clustering, etc. We refer to Wang & Zhang (2012) for a comprehensive review.

In the deep learning era, as a linear model, NMF is gradually replaced with deep neural networks (DNNs) with better feature extraction ability. Following this trend, various deep NMF methods have been proposed, where they replace the linear embedding layer with a multi-layer (optionally nonlinear) encoder, and enforce the non-negativity constraint by either architectural modules or regularization objectives. See Chen et al. (2021) for an overview of deep NMF methods. Nevertheless, these deep NMF methods can hardly achieve comparable performance to existing supervised and self-supervised learning methods for representation learning.

Our non-negative contrastive learning can also be regarded as a deep NMF method, as we also leverage a neural encoder and show the equivalence between the NCL objective and the NMF objective.

---

[3]In order to make this equivalence hold, Ding et al. (2005) add orthogonal constraints to NMF features, which are automatically satisfied under our one-hot latent label setting (Theorem 4).

[4]We note that theoretically, the K-means here is computed over the exponentially large population data $\mathcal{X}$. So it is not feasible in practice to utilize this connection to produce an NCL solution using CL features.

However, the key difference is that we do not apply NMF to the actual input (*e.g.,* the image), but to the co-occurrence matrix *implicitly* defined by data augmentation. In this way, we can transform the original NMF problem to a sampling-based contrastive loss, which allows us to solve it with deep neural networks.

## B PROOFS

### B.1 PROOF OF THEOROEM 2

*Proof.* We expend $\mathcal{L}_{\mathrm{NMF}}(F)$ and obtain:

$$
\begin{aligned}
\mathcal{L}_{\mathrm{NMF}}(F) =& \|\bar{A} - F_+ F_+^\top\|^2 \\
=& \sum_{x,x_+} \left( \frac{\mathcal{P}(x,x_+)}{\sqrt{\mathcal{P}(x)\mathcal{P}(x_+)}} - \sqrt{\mathcal{P}(x)} f_+(x)^\top \sqrt{\mathcal{P}(x_+)} f_+(x_+) \right)^2 \\
=& \sum_{x,x_+} \left( \frac{\mathcal{P}(x,x_+)^2}{\mathcal{P}(x)\mathcal{P}(x_+)} - 2\mathcal{P}(x,x_+) f_+(x)^\top f_+(x_+) \right. \\
& \left. + \mathcal{P}(x)\mathcal{P}(x_+) \left( f_+(x)^\top f_+(x_+) \right)^2 \right) \\
=& \underbrace{\sum_{x,x_+} \left( \frac{\mathcal{P}(x,x_+)^2}{\mathcal{P}(x)\mathcal{P}(x_+)} \right)}_{const} -2\mathbb{E}_{x,x_+} f_+(x)^\top f_+(x^+) + \mathbb{E}_{x,x^-} \left( f_+(x)^\top f_+(x^-) \right)^2 \\
=& \mathcal{L}_{\mathrm{NCL}} + const,
\end{aligned}
$$

which completes the proof. $\square$

### B.2 PROOF OF THEOREM 3

*Proof.* We expand $\mathcal{L}_{\mathrm{NMF}}(\phi)$ and obtain:

$$
\begin{aligned}
\mathcal{L}_{\mathrm{NMF}}(\phi) =& \sum_{x,x_+} \left( \frac{\mathcal{P}(x,x_+)}{\sqrt{\mathcal{P}(x)\mathcal{P}(x_+)}} - \sqrt{\mathcal{P}(x)} \phi(x)^\top \sqrt{\mathcal{P}(x_+)} \phi(x_+) \right)^2 \\
=& \sum_{x,x_+} \left( \frac{\mathcal{P}(x,x_+)}{\sqrt{\mathcal{P}(x)\mathcal{P}(x_+)}} - \sum_{i=1}^m \frac{\mathcal{P}(x)\mathcal{P}(x^+)\mathcal{P}(\pi_i|x)\mathcal{P}(\pi_i|x^+)}{\sqrt{\mathcal{P}(x)\mathcal{P}(x^+)}\mathcal{P}(\pi_i)} \right)^2 \\
=& \sum_{x,x_+} \left( \frac{\mathcal{P}(x,x_+)}{\sqrt{\mathcal{P}(x)\mathcal{P}(x_+)}} - \sum_{i=1}^m \frac{\mathcal{P}(\pi_i,x)\mathcal{P}(\pi_i,x^+)}{\sqrt{\mathcal{P}(x)\mathcal{P}(x^+)}\mathcal{P}(\pi_i)} \right)^2 \\
=& \sum_{x,x_+} \left( \frac{\mathcal{P}(x,x_+)}{\sqrt{\mathcal{P}(x)\mathcal{P}(x_+)}} - \sum_{i=1}^m \frac{\mathcal{P}(\pi_i)\mathcal{P}(x|\pi_i)\mathcal{P}(x^+|\pi_i)}{\sqrt{\mathcal{P}(x)\mathcal{P}(x^+)}} \right)^2 \\
=& \sum_{x,x_+} \left( \frac{\mathcal{P}(x,x_+)}{\sqrt{\mathcal{P}(x)\mathcal{P}(x_+)}} - \frac{\mathbb{E}_c \mathcal{P}(x|c)\mathcal{P}(x^+|c)}{\sqrt{\mathcal{P}(x)\mathcal{P}(x^+)}} \right)^2
\end{aligned}
$$

With Assumption 1, we know that $\mathcal{P}(x,x^+) = \mathbb{E}_c \mathcal{P}(x|c)\mathcal{P}(x^+|c)$. So $\mathcal{L}_{\mathrm{NMF}}(\phi) = 0$, *i.e.,* $\phi \in \arg\min \mathcal{L}_{\mathrm{NMF}}$. Combing with the equivalence between NMF and NCL, we obtain that $\phi$ is an optimal solution of $\mathcal{L}_{\mathrm{NCL}}$. $\square$

### B.3 PROOF OF THEOREM 4

*Proof.* When each sample only belongs to one latent class $c = \mu(x)$, we have $[\mathcal{P}(\pi_1|x), \cdots, \mathcal{P}(\pi_m|x)] = \mathbf{1}_{\mu(x)}$. Combined with Theorem 3, we have $\phi(x) = \sqrt{\frac{1}{\mathcal{P}(\pi_{\mu(x)})}} \mathbf{1}_{\mu(x)}$.

As there only exists one non-zero elements for $\phi(x)$, we obtain $\|\phi(x)\|_0 = 1$. And for the orthogonality, we have $\mathbb{E}_x \phi_i(x)\phi_j(x) = \sum_x \mathcal{P}(x)\sqrt{\frac{1}{\mathcal{P}(\pi_j)\mathcal{P}(\pi_j)}}\mathbb{1}_{\mu(x)=\pi_i}\mathbb{1}_{\mu(x)=\pi_j}$. So when $i \neq j$, $\mathbb{E}_x \phi_i(x)\phi_j(x) = 0$. And when $i = j$, $\mathbb{E}_x \phi_i(x)\phi_j(x) = \frac{1}{\mathcal{P}(\pi_i)}\sum_x \mathcal{P}(x)\mathbb{1}_{\mu(x)=\pi_i} = 1$. Then we obtain: $\mathbb{E}_x \phi(x)\phi(x)^\top = I$. $\qquad\square$

### B.4 PROOF OF THEOREM 5

*Proof.* First, we introduce two useful results from previous works.

**Lemma 1** (Lemma 1 of Paul & Chen (2016)). *For any $N \times N$ symmetric matrix $\bar{A}$, if $rank(\bar{A}) = k \leq N$, then the order $k$ exact symmetric NMF of $\bar{A}$ is unique up to an orthogonal matrix.*

**Lemma 2** (Lemma 1.1 of Minc (1988)). *The inverse of a non-negative matrix matrix $M$ is non-negative if and only if $M$ is a generalized permutation matrix.*

Let $\Phi \in \mathbb{R}_+^{N \times k}$ be the feature matrix composed of $N$ features, *i.e.*, the $x$-the row is $\sqrt{\mathcal{P}(x)}\phi(x)^\top$. Note that $\Phi$ satisfies $\bar{A} = \Phi\Phi^\top$, an exact non-negative matrix factorization of $\bar{A}$ (Theorem 4). If $\phi$ is not unique, there exists another $F_+ \neq \Phi$ and $F_+ \in \mathbb{R}_+^{N \times k}$ such that $\bar{A} = F_+ F_+^\top$.

According to Lemma 1, we can deduce $H = \Phi T$, where $T \in \mathbb{R}^{k \times k}$ is an orthogonal matrix. As $\Phi T$ is also an exact symmetric NMF of $\bar{A}$, it satisfies $\Phi T \geq 0$. For each $x \in \mathcal{X}$, it holds that $\phi(x)^\top T \geq 0$. Since for every latent class $c \in \mathcal{C}$, there exists a sample $x_c \in \mathcal{X}$ such that $P(c|x_c) = 1$, the $c$-th entry of $\phi(x_c)$ is $\sqrt{\frac{1}{p(\pi_c)}}$, while all other entries are 0. Denote the row vectors of matrix $T$ as $T_1, T_2, \ldots, T_k$. As $\phi(x_c)^\top T \geq 0$, we must have $T_c \geq 0$. Applying this deduction to every latent class, we have $\forall c \in \mathcal{C}, T_c \geq 0$, *i.e.*, $T \geq 0$.

Recall that $T \geq 0$ is also an orthogonal matrix with $T^\top T = I$. According to Lemma 2, $T$ must be a permutation matrix. Therefore, $\phi(\cdot)$ is unique under permutation. $\qquad\square$

### B.5 PROOF OF THEOREM 6

*Proof.* We consider the $y$-th dimension of the prediction:

$$
\begin{aligned}
&g_y^\star(\phi(x)) \\
=&(W^\star)^\top \phi(x) \\
=&\sum_{j=1}^m \sqrt{\mathcal{P}(\pi_j)\mathbb{1}_{\pi_j \in \mathcal{C}_y}}\frac{\mathcal{P}(\pi_j|x)}{\sqrt{\mathcal{P}(\pi_j)}} \\
=&\sum_{j=1}^m \mathcal{P}(\pi_j|x)\mathbb{1}_{\pi_j \in \mathcal{C}_y} \\
=&\mathcal{P}(y|x).
\end{aligned}
$$

So $\arg\max g^\star(\phi(x)) = \arg\max \mathcal{P}(y|x)$. In other words, the classifier attains the Bayes optimal classifier. $\qquad\square$

## C EXTENSION TO MULTI-MODAL CONTRASTIVE LEARNING

In the main paper, we have proposed non-negative contrastive learning (NCL) for self-supervised learning, where the co-occurrence matrix $A$ is symmetric by definition. Apart from the self-supervised setting, we know that contrastive learning is also successfully applied to visual-language representation learning that involves multi-modal data, *e.g.*, CLIP (Radford et al., 2021b). Zhang et al. (2023a) show that multi-modal contrastive learning is mathematically equivalent to an asymmetric matrix factorization problem. Based on this link, we can develop the asymmetric version of non-negative contrastive learning in the multi-modal setting, that can also enable the interpretability of visual-language representations.

Following the similar spirit of symmetric contrastive learning, asymmetric contrastive learning also aligns positive pairs together while pushing negative samples apart. However, there exist differences in the data sampling process. Taking the image-text pairs as an example, we use $X_V$ to denote the set of all visual data with distribution $\mathcal{P}_V$, and $\mathcal{X}_L$ to denote the set of all language data with distribution $\mathcal{P}_L$. Then the positive pairs $(x_v, x_l)$ are sampled from the joint multi-modal distribution $\mathcal{P}_M$ and the negative pairs $(x_v^-, x_l^-)$ are independently drawn from $\mathcal{P}_V$ nad $\mathcal{P}_L$. And we denote $A_M : (A_M)_{x_v, x_l} = \mathcal{P}_M(x_v, x_l)$ as the asymmetric co-occurrence matrix. For the ease of theoretical analysis, we consider the multi-modal spectral contrastive loss:

$$\mathcal{L}_{\text{MMCL}}(f_V, f_L) = -2\mathbb{E}_{x_v, x_l} f_V(x_v)^\top f_L(x_l) + \mathbb{E}_{x_v^-, x_l^-}(f_V(x_v^-)^\top f_L(x_l^-))^2, \qquad (11)$$

where $f_V$ and $f_L$ are two encoders that respectively encode vision and language samples. Correspondingly, we propose the multi-modal variant of NCL loss:

$$\mathcal{L}_{\text{MMNCL}}(f_{V+}, f_{L+}) = -2\mathbb{E}_{x_v, x_l} f_{V+}(x_v)^\top f_{L+}(x_l) + \mathbb{E}_{x_v^-, x_l^-}(f_{V+}(x_v^-)^\top f_{L+}(x_l^-))^2, \qquad (12)$$
$$\text{such that } f_{V+}(x_v) \geq 0, f_{L+}(x_l) \geq 0, \forall x_v \in \mathcal{X}_V, x_l \in \mathcal{X}_L.$$

And we introduce asymmetric non-negative matrix factorization objective:

$$L_{\text{ANMF}} = \|\bar{A}_M - F_{V+}F_{L+}^\top\|^2, \text{ such that } F_{V+}, F_{L+} \geq 0, \qquad (13)$$

where $(F_{V+})_{x_v} = \sqrt{\mathcal{P}_V(x_v)}f_{V+}(x_v)^\top$, $(F_{L+})_{x_l} = \sqrt{\mathcal{P}_L(x_l)}f_{L+}(x_l)^\top$ and $\bar{A}_M$ is the normalized co-occurrence matrix. Then we establish the equivalence between MMNCL and asymmetric non-negative matrix factorization objective following the proofs in Theorem 2:

*Proof.*

$$\mathcal{L}_{\text{ANMF}}(F_V, F_L) = \|\bar{A}_M - F_{V+}F_{L+}^\top\|^2$$

$$= \sum_{x_v, x_l} \left( \frac{\mathcal{P}_M(x_v, x_l)}{\sqrt{\mathcal{P}_V(x_v)\mathcal{P}_L(x_l)}} - \sqrt{\mathcal{P}_V(x_v)}f_{V+}(x_v)^\top \sqrt{\mathcal{P}_L(x_l)}f_{L+}(x_l) \right)^2$$

$$= \sum_{x_v, x_l} \left( \frac{\mathcal{P}_M(x_v, x_l)^2}{\mathcal{P}_V(x_v)\mathcal{P}_L(x_l)} - 2\mathcal{P}_M(x_v, x_l)f_{V+}(x_v)^\top f_{L+}(x_L) \right.$$

$$\left. + \mathcal{P}_V(x_v)\mathcal{P}_L(x_l)\left(f_{V+}(x_v)^\top f_{L+}(x_L)\right)^2 \right)$$

$$= \underbrace{\sum_{x_v, x_l} \left( \frac{\mathcal{P}_M(x_v, x_l)^2}{\mathcal{P}_V(x_v)\mathcal{P}_L(x_l)} \right)}_{const} - 2\mathbb{E}_{x_v, x_l} f_{V+}(x_v)^\top f_{L+}(x_l) + \mathbb{E}_{x_v^-, x_l^-}\left(f_{V+}(x_v^-)^\top f_{L+}(x_l^-)\right)^2$$

$$= \mathcal{L}_{\text{MMNCL}} + const.$$

$\square$

Following the proofs of Theorem 3, then we can obtain the optimal solutions of MMNCL under the positive generation assumption ($\forall x_v \in \mathcal{X}_V, x_l \in \mathcal{X}_L, \mathcal{P}_M(x_v, x_l) = \mathbb{E}_c \mathcal{P}(x_v|c)\mathcal{P}(x_l|c)$):

$$\phi_V(x) = \left[ \frac{1}{\sqrt{\mathcal{P}(\pi_1)}}\mathcal{P}(\pi_1|x_v), \ldots, \frac{1}{\sqrt{\mathcal{P}(\pi_m)}}\mathcal{P}(\pi_m|x_v) \right] \in \mathbb{R}_+^m, \forall\, x_v \in \mathcal{X}_V,$$

$$\phi_L(x) = \left[ \frac{1}{\sqrt{\mathcal{P}(\pi_1)}}\mathcal{P}(\pi_1|x_l), \ldots, \frac{1}{\sqrt{\mathcal{P}(\pi_m)}}\mathcal{P}(\pi_m|x_l) \right] \in \mathbb{R}_+^m, \forall\, x_l \in \mathcal{X}_L.$$

We observe that the optimal representations of MMNCL have a similar form to NCL, As a result, we can directly follow the theoretical analysis in NCL and prove that the learned representations of MMNCL also show semantic consistency such as dimensional clustering, sparsity and orthogonality.

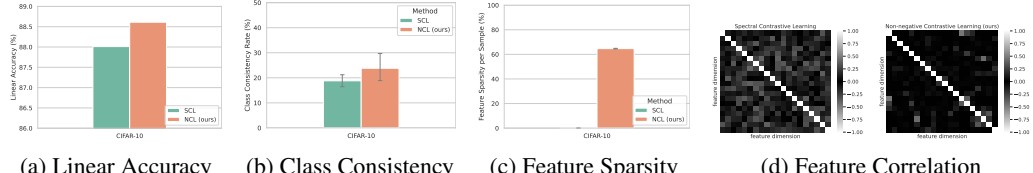

| (a) Linear Accuracy | (b) Class Consistency | (c) Feature Sparsity | (d) Feature Correlation |

Figure 5: Comparison between spectral contrastive learning (SCL) and non-negative contrastive learning (NCL) on CIFAR-10: a) linear probing accuracy; b) class consistency rate, measuring the proportion of activated samples that belong to their most frequent class along each feature dimension; c) feature sparsity, the average proportion of zero elements ($|x| < 1e^{-5}$) in the features of each test sample; d) dimensional correlation matrix $C$ of 20 random features:

$$\forall (i,j), C_{ij} = \mathbb{E}_x \tilde{f}_i(x)^\top \tilde{f}_j(x), \text{ where } \tilde{f}_i(x) = f_i(x) / \sqrt{\sum_x \left(f_i(x)\right)^2}.$$

## D    ADDITIONAL EMPIRICAL ANALYSES

### D.1    PERFORMANCE ON THE SPECTRAL CONTRASTIVE LEARNING LOSS

In the main text, we mainly consider the widely adopted InfoNCE loss. Here, we provide the results with the spectral contrastive loss (SCL) that our analysis is originally based on.

**Setup.** During the pretraining process, we utilize ResNet-18 as the backbone and train the models on CIFAR-10. We respectively pretrain the model with Spectral Contrastive Learning (SCL) and NCL for 200 epochs. We pretrain the models with batch size 256 and weight decay 0.0001. When implementing NCL, we follow the default settings of SCL. During the linear evaluation, we train a classifier following the frozen backbone for 100 epochs. For the experiments that compare semantic consistency, feature sparsity and feature correlation, we follow the settings in Section 3.1.

**Results.** As shown in Figure 5, NCL can also attain slight improvement over SCL in linear probing (88.6% *v.s.* 88.0%), while bring clear interpretability improvements in terms of class consistency (Figure 5b), sparsity (Figure 5c) and correlation (Figure 5d). It shows that NCL indeed works well across different CL objectives.

### D.2    COMPARISON TO SPARSITY REGULARIZATION

Another classic method to encourage feature sparsity is directly adding $\ell_1$ regularization on the output features, leading to the following regularized CL objective,

$$\mathcal{L}_{\text{CL}} + \lambda \mathbb{E}_x \| f(x) \|_1, \tag{14}$$

where $\lambda$ is a coefficient for regularization strength. We find that $\lambda = 0.01$ is a good tradeoff between accuracy and sparsity on CIFAR-10.

**Results.** As shown in Table 5, NCL outperforms $\ell_1$ regularized SCL on both classification accuracy (88.5% *v.s.* 87.2%), which indicates that $\ell_1$ regularization hurts downstream performance while our non-negative constraint can help improve it. As for feature sparsity, we find that 69.654% of NCL features are sparse while $\ell_1$ regularization only achieves 0.03% sparsity. Therefore, NCL is more effective than sparsity regularization in terms of both downstream performance and promoting feature sparsity.

Table 5: Comparing NCL to $\ell_1$-based sparsity regularization on CIFAR-10.

|  | Linear Accuracy (Encoder) (%) | Linear Accuracy (Projector) (%) | Feature Sparsity (%) |
| --- | --- | --- | --- |
| SCL | 88.0 | 85.4 | 0.00 |
| SCL + $\ell_1$ reg | 87.2 | 84.1 | 0.03 |
| **NCL** | **88.6** | **86.2** | **69.65** |

# E    EMPIRICAL VERIFICATIONS OF THEORETICAL ANALYSIS ON NCL

## E.1    EMPIRICAL VERIFICATIONS OF OPTIMAL REPRESENTATIONS

To further verify the theoretical analysis of the optimal solutions in Section 4, we further investigate properties of the learned representations on real-world datasets, *e.g.,* CIFAR-10. As we have no knowledge of the latent classes of CIFAR-10, we consider a simplified case, where we assume that the 10 classes in CIFAR-10 are latent classes, and generate positive samples by randomly sampling two augmented samples from the same class, in other words, supervised contrastive learning (Sup-Con) (Khosla et al., 2020). Then we train a ResNet-18 with CL and NCL objectives for 200 epochs, respectively.

By observing the largest 50 expected activation (EA) values of learned representations (Figure 6a), we find that the features of NCL are mostly activated in the first 10 dimensions, which is highly consistent with the ground-truth number of (latent) class in this setting with $C = 10$. In comparison, CL features do not differ much from one feature to another, showing that they do have good dimensional feature interpretability. Also, the eigenvalue distribution in Figure 6b reveals that CL features collapse even quicker than NCL features, and degrade to small values even before 10 dimensions, while the eigenvalues of NCL features only degrade quickly after the first 10 dimensions. Therefore, we believe that the learned NCL features do lie closer to the ideal ground truth.

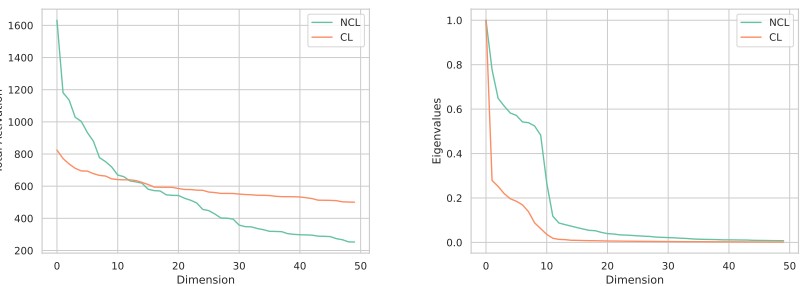

(a) Distribution of the 50 largest EA values    (b) Distribution of the 50 largest eigenvalues

Figure 6: Analysis of the learned features of CL and NCL on CIFAR-10.

## E.2    EMPIRICAL VERIFICATIONS OF ASSUMPTIONS IN THEOREM 5

To verify the assumptions we used in Theorem 5 (for each latent class, there exists at least one sample in the datasets only has that feature activated), we conduct experiments to observe the activated dimensions of different samples. As shown in Figure 7, samples generally have much fewer activated dimensions under NCL: around 70% examples have less than 20 activated dimensions, and in terms of the smallest, 16 examples are only activated on 2 dimensions. Considering that the noisy training in practice can lead to some redundant features and some gaps to the optimal solution, we believe that our assumption is indeed approximately true.

# F    ADDTIONAL RESULTS OF FEATURE PROPERTIES IN NCL

## F.1    DISENTANGLEMENT EVALUATION ON SYNTHETIC DATA

In the main paper, we evaluate feature disentanglement of CL and NCL on real-world datasets, which are difficult without knowledge of ground truth. Here, following the common practice of disentanglement literature, we also evaluate disentanglement on Dsprites, a toy dataset where we are aware of the ground-truth latents (Abdi et al., 2019). To be specific, we respectively train the models with original contrastive loss and NCL loss on Dsprites following the settings of Abdi et al. (2019). Then we evaluate the disentanglement by the widely adopted disentanglement metric Mutual Information Gap (MIG) (Chen et al., 2018).

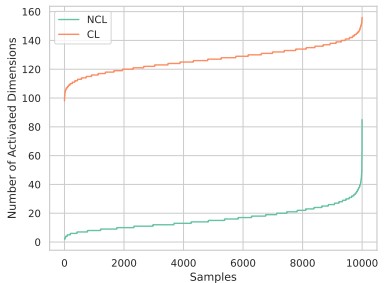

Figure 7: The number of activated dimensions of the representations learned by CL and NCL on CIFAR-10.

As shown in Table 6, NCL features also show much better disentanglement than CL on Dsprites as measured by MIG, which further verifies the advantages of NCL. Since Dspirtes is a toy model that is very different from real-world images, the default SimCLR augmentation could be suboptimal to obtain good positive samples. We leave more finetuning of NCL on these toy datasets for future work.

Table 6: Comparing disentanglement between CL and NCL on Dsprites.

| Method | MIG |
|--------|-------|
| CL     | 0.037 |
| **NCL** | **0.065** |

### F.2 More Interpretability Visualization

Besides the CIFAR-10 examples in Figure 1, we perform the interoperability visualization experiments on CIFAR-100 (Figure 8) and ImageNet-100 (Figure 9). We can still observe a clear distinction in semantic consistency: activated samples along each feature dimension often come from different classes in CL while those in NCL mostly belong to the same class. It verifies that NCL has clear advantages over CL in feature interpretability on multiple datasets.

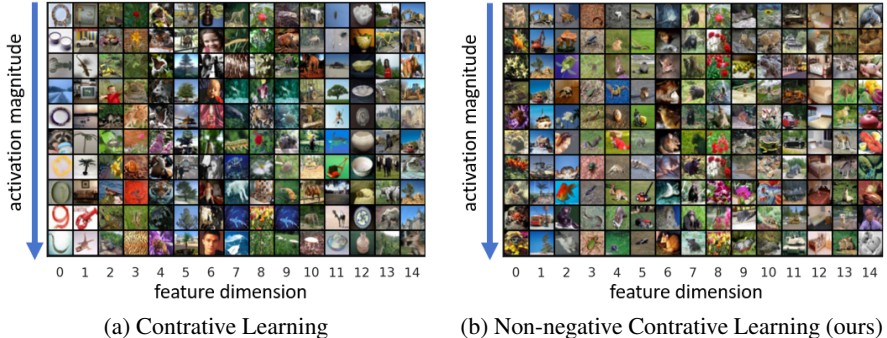

(a) Contrative Learning      (b) Non-negative Contrative Learning (ours)

Figure 8: Visualization of CIFAR-100 test samples with the largest values along each feature dimension (sorted according to activation values, see Appendix G).

### F.3 Performance under Dynamic Feature Length

To further investigate the benefit of sparsity in the representations learned by NCL, we conduct additional experiments to observe the performance of expected activation (EA) with different numbers of selected dimensions. As shown in Figure 10, EA-based NCL features keep a high level of mAP

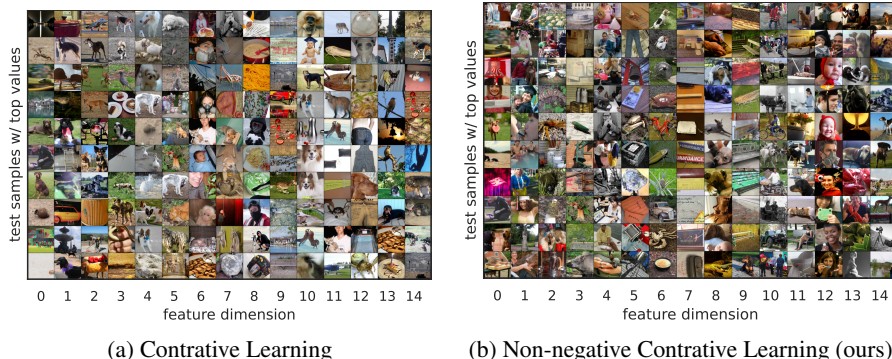

(a) Contrative Learning        (b) Non-negative Contrative Learning (ours)

Figure 9: Visualization of ImageNet-100 test samples with the largest values along each feature dimension (sorted according to activation values, see Appendix G).

and have almost no degradation of performance. Instead, CL features keep degrading under fewer features and underperform NCL a lot at last. This experiment shows the clear advantage of the sparsity and disentanglement properties of NCL.

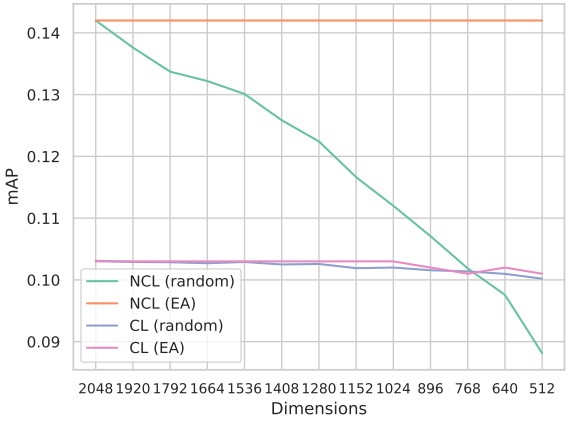

Figure 10: Image retrieval with CL and NCL features and two feature selection criteria (random, EA) on ImageNet-100.

### F.4 DOWNSTREAM PERFORMANCE OF PROJECTOR OUTPUT

Following the common practice (Chen et al., 2020; He et al., 2020), in the main paper, feature transferability is by default evaluated based on the encoder features (before projector) to attain better downstream performance. For completeness, we also evaluate the features of projector outputs on CIFAR-10 and CIFAR-100. As shown in Table 7, we can see that NCL (w/ ReLU) output can achieve comparable performance to CL outputs (w/o ReLU) while providing much better feature interpretability as shown in Figure 3. Besides, Tables 3 & 3b shows that when pursuing interpretability, NCL can also induce better encoder features than CL. Overall, we believe that NCL is indeed favorable since it attains better feature interpretability while maintaining feature usefulness.

## G DETAILS OF EXPERIMENT SETUP

### G.1 IMPLEMENTATION AND EVALUATION OF NON-NEGATIVE FEATURES

In a standard contrastive learning method like SimCLR, the encoder is composed of two subsequential components: the backbone network (*e.g.,* a ResNet) $g$, and the projector network (usually an

Table 7: Downstream classification performance (accuracy in percentage) with projector outputs.

| Method | CIFAR-10 | | CIFAR-100 | |
|---|---|---|---|---|
| | LP | FT | LP | FT |
| CL (w/o ReLU) | 85.6±0.1 | 92.1±0.1 | 53.6±0.3 | 71.4±0.2 |
| NCL (w/o ReLU) | 85.4±0.3 | 92.4±0.1 | 53.7±0.2 | 71.5±0.2 |

MLP network) $p$. The projector output $f(x) = p(g(x))$ is then used to compute the InfoNCE, using the cosine similarity between positive and negative features. We apply the reparameterization at the final output of the projector output, *i.e.,* $f_+(x) = \sigma_+(f(x))$. Then, the non-negative feature is fed into computing the InfoNCE loss. All hyperparameters stay the same as the default ones.

**Evaluation.** Since non-negative constraints are only imposed on the projector output, we regard the projector outputs as the learned features in the visualization, feature selection, and feature disentanglement experiments, which aligns well with our theoretical analysis. When evaluating the feature transferability to downstream classification, following the convention, we also discard the projector and use the backbone network output for linear probing and fine-tuning.

## G.2 EXPERIMENT DETAILS OF VISUALIZATION RESULTS AND PILOT STUDY

In all experiments in this part, we adopt SimCLR as the default backbone method (with ResNet-18), train the models for 200 epochs, and collect results on CIFAR-10 test data. In Figure 1, we sort feature dimensions according to expected activation (taking absolute values for contrastive learning features), and select the top samples with the largest activation in each dimension. In other results, the feature dimensions and samples are all randomly selected (without cherry-picking). Dimensions that have zero activation on all test samples (*i.e.,* dead features) are excluded.

## G.3 EXPERIMENT DETAILS OF FEATURE DISENTANGLEMENT

In practice, for the ease of computation, we first decompose the SEPIN metric based on the definition of the mutual information:

$$
\begin{aligned}
I(x, f_{r_i}(x)|f_{\neq r_i}(x)) &= I(x, (f_{r_i}(x), f_{\neq r_i}(x))) - I(x, f_{\neq r_i}(x)) \\
&= I(x, f(x)) - I(x, f_{\neq r_i}(x))
\end{aligned}
\tag{15}
$$

In the next step, we calculate two terms respectively. As it is quite hard to calculate the mutual information directly, we use a tractable lower bound: the InfoNCE (Oord et al., 2018) to estimate that. To be specific, for the mutual information $I(x, f(x))$, we estimate it with $\mathcal{L}_{\mathrm{NCE}}(f)$, *i.e.,*

$$
I(x, f(x)) \approx \mathcal{L}_{\mathrm{NCE}}(f) = -\mathbb{E}_{x,x^+} \log \frac{\exp(f(x)^\top f(x^+))}{\exp(f(x)^\top f(x_+)) + \frac{1}{M}\sum_{i=1}^{M} \exp(f(x)^\top f(x_i^-))},
\tag{16}
$$

where $f$ is the learned encoder. Similarly, for the mutual information $I(x, f_{\neq i}(x))$, we obtain:

$$
I(x, f_{\neq i}(x)) \approx \mathcal{L}_{\mathrm{NCE}}(f_{\neq i}) = -\mathbb{E}_{x,x^+} \log \frac{\exp(f_{\neq i}(x)^\top f_{\neq i}(x^+))}{\exp(f_{\neq i}(x)^\top f_{\neq i}(x^+)) + \frac{1}{M}\sum_{i=1}^{M} \exp(f_{\neq i}(x)^\top f_{\neq i}(x_i^-))}.
\tag{17}
$$

Then we respectively calculate two terms on the test data of ImageNet-100 with the encoders learned by NCL and CL. Following the definition of SEPIN@$k$, we sort the dimensions based on the values of SEPIN, and SEPIN@$k$ calculates the mean values of $k$ largest SEPIN values of learned representations.

## G.4 EXPERIMENT DETAILS OF FEATURE SELECTION

During the pretraining process, we utilize ResNet-18 (He et al., 2016) as the backbone and train the models on ImageNet-100 (Deng et al., 2009). We follow the default settings of SimCLR (Chen et al., 2020) and pretrain the model for 200 epochs.

**Linear Probing on Selected Dimensions.** During the evaluation process, we train a linear classifier following the frozen representations with the default settings of linear probing in SimCLR. As shown

in Table 1, the features of NCL selected with EA show significantly better linear accuracy, which verifies that the features of NCL are interpretable and EA values can discover the most important semantic dimensions related to the ground-truth labels.

**Image Retrieval on Selected Dimensions.** For each sample, we first encode it with the pretrained networks and then select dimensions from the features. Then we find 10 images that have the largest cosine similarity with the query image and calculate the mean average precision @ 10 (mAP@10) that returned images belong to the same class as the query ones. As shown in Table 1, we observe the dimensions selected by EA values of NCL can find the intra-class neighbors more accurately, which further shows that NCL can find the most important features that are useful in downstream tasks.

**Transfer Learning on Selected Dimensions.** In the downstream transfer learning tasks, we train a linear classifier on stylized ImageNet-100 (Geirhos et al., 2018) following the frozen representations with the default settings of linear probing. As shown in Table 1, the 512 dimensions with the largest EA values show superior performance, which implies that the estimated importance of NCL is robust and effective in downstream transfer learning tasks.

### G.5    EXPERIMENT DETAILS OF DOWNSTREAM CLASSIFICATION

During the pretraining process, we utilize ResNet-18 (He et al., 2016) as the backbone and train the models on CIFAR-10, CIFAR-100 and ImageNet-100 (Deng et al., 2009). We pretrain the model for 200 epochs on CIFAR-10, CIFAR-100, and 100 epochs for ImageNet-100. We compare NCL and the original contrastive learning with the SimCLR (Chen et al., 2020) backbone. For CIFAR-10 and CIFAR-100, the projector is a two-layer MLP with hidden dimension 2048 and output dimension 256. And for ImageNet-100, the projector is a two-layer MLP with hidden dimension 16384 and output dimension 2048. We pretrain the models with batch size 256 and weight decay 0.0001. When implementing NCL, we follow the default settings of SimCLR.

During the evaluation process, we consider three generalization tasks: in-distribution linear evaluation, in-distribution finetuning and out-of-distribution generalization. During the linear evaluation, we train a classifier following the frozen backbone pretrained by different methods for 50 epochs. For the in-distribution finetuning, we train both the backbone and the classifier for 30 epochs. And for the out-of-distribution generalization, we use the linear classifier obtained with the linear evaluation process on original ImageNet-100. Then we evaluate the linear accuracy on stylized ImageNet (Geirhos et al., 2018), ImageNet-Sketch (Wang et al., 2019), ImageNet-Corrpution (Hendrycks & Dietterich, 2019). As we pretrain the network on ImageNet-100, we select the samples of the corresponding 100 classes from these out-of-distribution datasets and evaluate the accuracy.

### G.6    EXPERIMENT DETAILS OF SUPERVISED LEARNING

For the fine-tuning tasks, we first train a model by SimCLR. During the pretraining process, we utilize ResNet-18 (He et al., 2016) as the backbone and train the models on ImageNet-100 (Deng et al., 2009). We pretrain the model for 200 epochs. We use a projector which is a two-layer MLP with hidden dimension 16384 and output dimension 2048. We pretrain the models with batch size 256 and weight decay 0.0001. During the fine-tuning process, we train a classifier following the backbone for 50 epochs respectively with supervised and non-negative supervised learning and we follow the default settings of fine-tuning. When implementing the non-negative supervised learning, we select the ReLU function as the non-negative operator.

For the training from scratch, we also follow the default settings of fine-tuning. And we training the randomly initialized ResNet-18 with supervised and non-negative supervised learning for 100 epochs on ImageNet-100. When implementing the non-negative supervised learning, we select the ReLU function as the non-negative operator. It is worth noting that we also apply ReLU functions on the linear classifiers during the training process.

In both tasks, we respectively apply the CE and NCE loss on the encoders with and without the projectors. We follow the default settings of the projectors used in SimCLR, i.e., the projector is a two-layer MLP with hidden dimension 16384 and output dimension 2048.

