# OpenReview forum: "Non-negative Contrastive Learning"
_ICLR.cc/2024/Conference — ICLR 2024 poster_

### Official Review · Reviewer_kpGZ · 2023-10-30

**Soundness:** 2 fair
**Presentation:** 3 good
**Contribution:** 3 good
**Rating:** 6
**Confidence:** 3

**Summary:**

The authors propose a novel contrastive learning method called "nonnegative contrastive learning", which takes its name from nonnegative matrix factorization. Using an output activation function for the learned features, e.g. $ReLU(f(\mathbf{x}))$, the authors make the connection of the nonnegative matrix factorization minimization loss to the spectral contrastive learning loss.
The novel objective should lead to more interpretable features.

**Strengths:**

- Interpretability of ML methods and learned representations is an important problem. For supervised and self-supervised learning. Therefore, I like the idea of the paper and its research direction.
- Combining contrastive learning and nonnegative matrix factorization is a cool idea and innovative.
- The paper is well written and a good read. Interesting!

**Weaknesses:**

- In my opinion, running experiments over multiple seeds and reporting the mean and standard deviation is essential and. necessary. For me, this is an important point. And to me, it has to be fulfilled to be accepted.
- The y-axes in Figure 5 give a biased/wrong impression of the performances of the different models, in my opinion. In combination with the lack of standard deviations, it is difficult to evaluate the performance differences of the two methods.
- I understand that evaluating disentanglement on real-world datasets is challenging given the lack of ground truth. Another possible way of really evaluating the disentanglement of the learned representations would have been the use of a toy dataset for this experiment. The known problems with estimating MI and not only a lower bound of it make it difficult to evaluate the results for this experiment.


### Comment
I really like the idea of the paper and the direction of research. Therefore, I am more than happy to change my score if the experimental evaluation and the reporting of results include some measure of their stability  (e.g. mean and standard deviations or something similar).

**Questions:**

- what is the connection/link between semantic consistency and clustering accuracy? I could not find any details on that in the manuscript.
- Is the CL objective also trained with the spectral loss function? The results in the original paper seem to indicate that it does not lead to the same performance as in the original SimCLR paper. Maybe it would make sense to include the results generated by the standard SimCLR objective.
- are there more interpretability results than the qualitative results provided in Fig. 1?
- In sec. 4.2 you describe the feature selection procedure based on the TA. Do the results hold if you take the sum over absolute feature values, e.g. $ TA_i (f) = \sum_x \tilde{f}_i (x)$? Given the nonnegative constraints of the NCL features, the metric seems more tailored towards NCL than CL.

---

> ### Author Response · Authors · 2023-11-19
> **Response to Reviewer kpGZ (1/2)**
>
> Thanks for your careful reading and critical review. We are very happy to hear that you *“really like the idea of the paper and the direction of research.”* Following your suggestions, we have added reported mean and std results across multiple runs, which also agree with with our analysis before. We further address each of your concerns below, and hope you find them satisfactory.
>
> ---
>
> **Q1**. In my opinion, running experiments over multiple seeds and reporting the mean and standard deviation is essential and. necessary. For me, this is an important point. And to me, it has to be fulfilled to be accepted.
>
> **A1**. Following your suggestion, we have replaced **all benchmark results** with mean and std values obtained from 3 random trials (see **updated Tables 1-4**). It can be seen that **NCL indeed outperforms CL significantly in most feature selection and downstream tasks.**
>
> ---
>
> **Q2**. The y-axes in Figure 5 give a biased/wrong impression of the performances of the different models, in my opinion. In combination with the lack of standard deviations, it is difficult to evaluate the performance differences of the two methods.
>
> **A2**. Thanks for pointing it out. To be clearer, we have replaced Figure 5 with **Table 2** to better illustrate the results, together with **error bars calculated over 3 random trials**. It can be seen that with the proposed selection criterion (TA with absolute values), NCL indeed outperforms CL significantly in most cases.
>
> ---
>
> **Q3**. I understand that evaluating disentanglement on real-world datasets is challenging given the lack of ground truth. Another possible way of really evaluating the disentanglement of the learned representations would have been the use of a toy dataset for this experiment. The known problems with estimating MI and not only a lower bound of it make it difficult to evaluate the results for this experiment.
>
> **A3**.  Following your suggestion, we conduct additional experiments on Dsprites, a toy dataset with ground truth latent vectors that is widely adopted in the disentanglement literature. We train a small CNN encoder with CL and NCL, respectively, and measure feature disentanglement with a common metric Mutual Information Gap (MIG) proposed in [1]. For simplicity, we adopt the default training configuration and augmentation strategy as in SimCLR.
>
> **Results.** The preliminary results are shown below. We can see that NCL features show much better disentanglement than CL (higher the better), which further shows the advantages of NCL. We have included this result in **Appendix F.3**. Since Dspirtes is a toy model that is very different from real-world images, the default SimCLR augmentation could be suboptimal to obtain good positive samples. We leave more finetuning of NCL on these toy datasets for future work.
>
> *Comparison of feature disentanglement on Dspirtes.*
>
> | Metric | CL | NCL |
> | --- | --- | --- |
> | MIG | 0.037 | 0.065 |
>
> [1] Chen, Ricky TQ, et al. "Isolating sources of disentanglement in variational autoencoders." In NeurIPS. 2018.
>
> ---
>
> **Q4**. What is the connection/link between semantic consistency and clustering accuracy? I could not find any details on that in the manuscript.
>
> **A4**. As defined in Figure 4a caption, semantic consistency measures belonging to the most frequent class $c$ activated in each dimension $x$. It means that if we assign all activated samples in dimension $x$ to class $c$ (as a super naive clustering method), the expected clustering accuracy is  semantic consistency. But indeed, this naive method may not fully reflect the true clustering accuracy. To be more rigorous, we have removed the lines discussing their connections in the revision. For a better measure of clustering performance, we believe the linear probing classification accuracy (Table 3) is a better surrogate.
>
> ---
>
> **Q5**. Is the CL objective also trained with the spectral loss function? The results in the original paper seem to indicate that it does not lead to the same performance as in the original SimCLR paper. Maybe it would make sense to include the results generated by the standard SimCLR objective.
>
> **A5**. As discussed in Sec 2.3, the NCL reparameterization trick also applies to other CL objectives, such as InfoNCE. Due to its wide popularity, we adopt the SimCLR framework (with **InfoNCE** loss) by default in our paper (mentioned at the beginning of Sec 4).
>
> For completeness, we also add the results of SCL loss in revision (**Appendix F.1**). As quoted in the following table, with the SCL loss, NCL also yields clear gains over CL on both feature interpretability and downstream performance.
>
> *Comparison of SCL and its NCL version on CIFAR-10.*
>
> |  | Linear Accuracy  | Semantic Consistency | Sparsity | Correlation Error |
> | --- | --- | --- | --- | --- |
> | SCL | 88.0 | 19.0 | 0.002 | 0.117 |
> | NCL | 88.6 | 24.0 | 64.654 | 0.004 |

---

> ### Author Response · Authors · 2023-11-19
> **Response to Reviewer (2/2)**
>
> **Q6**. Are there more interpretability results than the qualitative results provided in Fig. 1?
>
> **A6**. Yes, we have! On the one hand, during revision, we add more qualitative results on other datasets in Appendix F.2. On the other hand, in Fig 4a, we have designed the **Semantic Consistency  (SC) to measure the dimension-wise feature interpretability, which is exactly the quantitative version of Figure 1**. As described in Fig 4a caption, SC measures the proportion of activated samples that belong to their most frequent class along each feature dimension. Since Fig 4a shows that NCL obtains much higher Semantic Consistency, it means that **quantitatively, NCL indeed has more consistent semantics along each feature dimension**. Other than that, we also measure the **feature sparsity and correlation in Fig 4b and Fig 4c,** as another two desirable properties of interpretable features. Please let us know if there is more to clarify.
>
> ---
>
> **Q7**. In sec. 4.2 you describe the feature selection procedure based on the TA. Do the results hold if you take the sum over absolute feature values, e.g. $TA_i (f) = \sum_x \tilde{f}_i (x)$? Given the nonnegative constraints of the NCL features, the metric seems more tailored towards NCL than CL.
>
> **A7**. That is a good observation. In fact, we have also tried taking the sum over the raw feature values, and found that it leads to even worse performance on CL (table below). Thus, it seems that the absolute feature value is also a better indicator for feature importance in CL. In either case, NCL with TA (abs) can outperform CL by a large margin.
>
> | CIFAR-10 | Linear Probing |  | Image Retrieval |  | Transfer Learning |  |
> | --- | --- | --- | --- | --- | --- | --- |
> |  | CL | NCL | CL | NCL | CL | NCL |
> | all(512) | 67.2$\pm$0.2 | 68$\pm$0.2 | 10.3$\pm$0.1 | 11.4$\pm$0.2 | 27.5$\pm$0.1 | 30.6$\pm$0.1 |
> | Random (256) | 65.7$\pm$0.1 | 65.3$\pm$0.2 | 9.7$\pm$0.1 | 7.8$\pm$0.2 | 26.2$\pm$0.4 | 24.7$\pm$0.1 |
> | TA (256, raw) | 65.9$\pm$0.2 | 66.2$\pm$0.2 | 9.8$\pm$0.2 | 8.4$\pm$0.1 | 26.1$\pm$0.1 | 27.3$\pm$0.2 |
> | TA (256, abs) | 66.3$\pm$0.2 | **67.5$\pm$0.1** | 9.8$\pm$0.1 | **11.2$\pm$0.1** | 26.9$\pm$0.2 | **30.5$\pm$0.1** |
>
> ---
>
> Thank you again for your encouraging comments and valuable feedbacks. We have updated the paper following your suggestions, and we respectfully suggest that you could re-evaluate it based on these updated results. We are very happy to address your remaining concerns on our work.

---

> > ### Comment · Reviewer_kpGZ · 2023-11-21
> >
> > Dear authors,
> >
> > Thanks for your reply and your changes to the manuscript. Especially presenting results for three different random seeds.
> > I am happy to update my score.
> >
> > However, two questions remain regarding Table 2.
> > - it seems is not better in sorting important features compared to CL. Both methods have approximately the same decrease in performance compared to using all 512 features. NCL starts at a higher level and is able to maintain this performance gap.
> > - Why are the numbers different for NCL for $TA(raw)$ and $TA(abs)$? Shouldn't all features of the NCL method be non-negative given the ReLU output activation (eq. (6))?

---

> ### Author Response · Authors · 2023-11-22
> **Further Response to Reviewer kpGZ**
>
> Dear Reviewer kpGZ,
>
> Thanks for appreciating our response and for improving the score. We are happy that you find the updated results satisfactory. We will address your remains questions below.
>
> ---
>
> **Q8**. It seems is not better in sorting important features compared to CL. Both methods have approximately the same decrease in performance compared to using all 512 features. NCL starts at a higher level and is able to maintain this performance gap.
>
> | CIFAR-10 | Linear Probing |  | Image Retrieval |  | Transfer Learning |  |
> | --- | --- | --- | --- | --- | --- | --- |
> |  | CL | NCL | CL | NCL | CL | NCL |
> | all(512) | 67.2$\pm$0.2 | 68$\pm$0.2 | 10.3$\pm$0.1 | 11.4$\pm$0.2 | 27.5$\pm$0.1 | 30.6$\pm$0.1 |
> | Random (256) | 65.7$\pm$0.1 (-1.5) | 65.3$\pm$0.2 (-2.7) | 9.7$\pm$0.1 (-0.6) | 7.8$\pm$0.2 (-3.6) | 26.2$\pm$0.4 (-1.3) | 24.7$\pm$0.1 (-5.9) |
> | TA (256, abs) | 66.3$\pm$0.2 (-0.9)  | **67.5$\pm$0.1 (-0.5)** | 9.8$\pm$0.1 (-0.5) | **11.2$\pm$0.1 (-0.2)** | 26.9$\pm$0.2 (-0.6) | **30.5$\pm$0.1 (-0.1)** |
>
> **A8**. We are afraid that your observation might not be accurate enough. For a clear examination, we calculate the drop of mean accuracy, as shown above. We can see that with TA selection, **NCL always drops fewer than CL on all three tasks (-0.9 vs -0.5, -0.5 vs -0.2, and -0.6 vs -0.1)**. This result is more nontrivial considering the fact that **NCL drops much more than CL under random selection** (e.g., -0.6 (CL) vs -3.6 (NCL) on image retrieval). It shows that the TA criterion (designed based on feature disentanglement) is very useful for NCL features but not much helpful for CL features.
>
> To fully compare CL and NCL on feature selection, we add a new plot in **Figure 10** (Appendix F.8), showing that NCL features indeed have much fewer drops in accuracy when we gradually reduce the feature dimension from 512 to 256.
>
> This distinction is actually another evidence showing that NCL has better disentanglement. When features are disentangled, randomly dropping features may lose important information (like in NCL), while if features are entangled, these information can be inferred from other entangled dimensions (like in CL). The fact that TA selection can largely restore NCL’s performance from random selection shows that NCL features are indeed sorted well according to feature importance.
>
> ---
>
> **Q9**. Why are the numbers different for NCL for TA(raw) and TA(abs)? Shouldn't all features of the NCL method be non-negative given the ReLU output activation (eq. (6))?
>
> **A9**. Here, for a fair comparison, we do not apply ReLU transformation to both CL and NCL, which leads to the difference you mentioned. To avoid confusion, we have modified it to “TA (w/o ReLU)”, and TA (w/ ReLU).
>
> ---
>
> Hope the explanations above could address your concern! Please let us know if there is more to clarify.

---

### Official Review · Reviewer_eLwm · 2023-11-01

**Soundness:** 3 good
**Presentation:** 3 good
**Contribution:** 3 good
**Rating:** 6
**Confidence:** 3

**Summary:**

The paper proposes the use of non-negative transformations on learned features in contrastive learning, which it calls Non-negative Contrastive Learning (NCL). It shows that non-negativity improves feature interpretability, sparsity and orthogonality compared to traditional CL. For each of these properties, the paper derives relevant guarantees for the ideal features, under certain assumptions of the data generating process. The paper also includes the results of experiments to show that NCL is superior to CL in feature disentanglement, feature selection and downstream classification.

**Strengths:**

The paper derives several useful guarantees for NCL under ideal circumstances. Although I did not delve deep into the derivations, they seem sound based on the non-exhaustive checking I did.

The proposed benefits of NCL over CL are further backed up by the experimental results. Although the derivation are in-depth, the proposed modification to CL is fairly simple to understand and implement. Whilst some parts would benefit from a more thorough explanation, the paper is generally well-written.

**Weaknesses:**

A few sections required re-reading and/or reading other related papers to fully comprehend. For example, in Section 2.1 Preliminary on Contrastive Learning, the notation, the ideas about the natural data vs. augmentation data, population data vs. empirical data, positive samples and negative samples could be introduced more thoroughly, as is done in HaoChen et al. (2021). I appreciate that this may be due to lack of available space.

In section C.1, it is stated that the feature transferability to downstream classification is evaluated without using the projector. Does this mean that the features used are not non-negative? Section 4.3 claims that the downstream classification “aligns well with the common belief that disentangled features are more robust against domain shifts”, and yet the empirical and theoretical justification for the features being disentangled is based on the non-negative features (after the projector). This appears inconsistent.

The theoretical guarantees derived in the paper are based on assumptions. There is insufficient discussion of the applicability of these assumptions, and the guarantees, when real world data is used and the learned features are not ideal. For example, for each feature (representing a latent class), is there at least one sample in the data sets used that (approximately) only has that feature activated? Is learning something close to the ideal features reliant on a feasible amount of training data and a feasible amount of augmentations?

It is unclear to me why the feature selection task is chosen to show the sparsity of the NCL solution. Why is the sparsity evidenced by performance with a subset of the original features rather than a subset of the learned features?

In Appendix C.3, the derivation of lower bound of mutual information as the NCE loss should be more thorough, or cite a paper with a more closely aligned derivation, as it has a different form to the original paper (Oord et al., 2018). Also, the NCE loss does not include positive samples as it does in the main text. Is this intentional?

The expectation in equation 1 appears to be over just positive samples.

The paper would be more reproducible if the code were made public.

Comment: My initial rating is "5: marginally below the acceptance threshold", but if the points above are well-addressed, I would be willing to increase my rating.

**Questions:**

In section C.1, it is stated that the feature transferability to downstream classification is evaluated without using the projector. Does this mean that the features used are not non-negative?

For each feature (representing a latent class), is there at least one sample in the actual data sets used that (approximately) only has that feature activated (as in assumption of Theorem 5).

Is learning something close to the ideal features reliant on a feasible amount of training data and a feasible amount of augmentations?

Why is the sparsity evidenced by performance with a subset of the original features rather than a subset of the learned features?

In Appendix C.3, the NCE loss does not include positive samples as it does in the main text. Is this intentional?

---

> ### Author Response · Authors · 2023-11-19
> **Response to Reviewer eLwm (1/2)**
>
> Thanks for your detailed reading and appreciating the simplicity of our method. We address your main concerns as follows.
>
> ---
>
> **Q1.** In Section 2.1 Preliminary on Contrastive Learning, the notation, the ideas about the natural data vs. augmentation data, population data vs. empirical data, positive samples and negative samples could be introduced more thoroughly, as is done in HaoChen et al. (2021). I appreciate that this may be due to lack of available space.
>
> **A1**. Indeed, in the preliminary, we mainly introduce the population data version for simplicity. We have revised the writing to be clearer on introducing the natural and augmented data, and the positive and negative samples.
>
> ---
>
> **Q2**. In section C.1, it is stated that the feature transferability to downstream classification is evaluated without using the projector. Does this mean that the features used are not non-negative? Section 4.3 claims that the downstream classification “aligns well with the common belief that disentangled features are more robust against domain shifts”, and yet the empirical and theoretical justification for the features being disentangled is based on the non-negative features (after the projector). This appears inconsistent.
>
> **A2**. Yes. Following the common practice as in SimCLR, MoCo, etc, feature transferability is also evaluated based on the encoder features (before projector) to attain better downstream performance. Thus, it would be more precise to change the claim to *“[the downstream performance] shows that disentangled feature learning can yield representations that are more robust against domain shifts”*.
>
> For completeness, we also evaluate the features of projector outputs, and add the results to **Table 6** (**Appendix F.5**). As quoted below, we can see that NCL (w/ ReLU) output can achieve comparable performance to CL outputs (w/o ReLU) while **providing much better feature interpretability as shown in Figure 4**. Besides, Table 3 shows that when pursuing interpretability, NCL can also **induce better encoder features** than CL. Overall, we believe that NCL is indeed favorable since it attains better feature interpretability while maintaining feature usefulness.
>
> *Downstream performance with projector outputs.*
>
> |  | CIFAR-10 |  | CIFAR-100 |  |
> | --- | --- | --- | --- | --- |
> |  | LP | FT | LP | FT |
> | CL (w/o ReLU) | 85.6$\pm$0.1 | 92.1$\pm$0.1 | 53.6$\pm$0.3 | 71.4$\pm$0.2 |
> | NCL (w/ ReLU) | 85.4$\pm$0.3 | 92.4$\pm$0.1 | 53.7$\pm$0.2 | 71.5$\pm$0.2 |
>
> ---
>
> **Q3**. The theoretical guarantees derived in the paper are based on assumptions. There is insufficient discussion of the applicability of these assumptions, and the guarantees, when real world data is used and the learned features are not ideal. For example, for each feature (representing a latent class), is there at least one sample in the data sets used that (approximately) only has that feature activated?
>
> **A3**. That’s a good question! Since latent classes $C$ are not observable, it would be hard to actually verify the assumptions and the guarantees. We note that the original paper Arora et al. (2019) also uses intuitive justification for this assumption. Nevertheless, we can justify this assumption under some special cases, when we know their exact meanings:
>
> - **Latent Class = Observed Class.** When $C$ corresponds to the observed labels $\mathcal{Y}$ (i.e., supervised contrastive learning), it is easy to see that this assumption holds, as it only requires a very mild condition: for each class (e.g., dog), there is indeed one image that only contains a dog, and no objects from the other classes. It is easy to see that ImageNet and CIFAR datasets all fit this criterion.
> - **Latent Class =  Individual Sample**. If each sample represents a latent class, this assumption would require this example is not equivalent to any other samples in this dataset. This is also very mild since any dataset with no duplicated samples satisfies this condition.
>
> Therefore, when we know the ground truth labels, it is easy to see that our assumptions are very natural and are very likely to hold in practice.
>
> > Is learning something close to the ideal features reliant on a feasible amount of training data and a feasible amount of augmentations?
> >
>
> We note that practical NN training is highly stochastic, non-convex, and influenced by many factors (architectures, optimizers, augmentations, etc), so it is generally hard to attain the exact optimal solution. To examine the general trend, we consider the supervised contrastive learning setting, when we can observe the latent classes as the ground-truth labels. As shown in **Figure 6 in Appendix 4**, the learned features of CL and NCL have clear differences. In particular, **NCL features are more concentrated on the top 10 features**, which roughly aligns with our theoretical analysis that the optimal features are located at $C$ dimensions ($C=10$ for CIFAR-10).

---

> ### Author Response · Authors · 2023-11-19
> **Response to Reviewer eLwm (2/2)**
>
> **Q4**. It is unclear to me why the feature selection task is chosen to show the sparsity of the NCL solution. Why is the sparsity evidenced by performance with a subset of the original features rather than a subset of the learned features?
>
> **A4**. As mentioned in Section 4.2, we explore *“utilizing this property for selecting a few important feature dimensions among the learned representations". Thus*, the feature selection task indeed refers to **selecting a subset of the learned features**, instead of from the original input features.
>
> ---
>
> **Q5**. In Appendix C.3, the derivation of lower bound of mutual information as the NCE loss should be more thorough, or cite a paper with a more closely aligned derivation, as it has a different form to the original paper (Oord et al., 2018). Also, the NCE loss does not include positive samples as it does in the main text. Is this intentional?
>
> **A5**. Thanks for pointing it out, this is a typo. We have fixed it in the revision.
>
> ---
>
> **Q6**. The expectation in equation 1 appears to be over just positive samples.
>
> **A6**. Thanks for pointing it out. We have added the negative samples in the revision.
>
> ---
>
> **Q7**. The paper would be more reproducible if the code were made public.
>
> **A7**. We will definitely publish the source code if accepted.
>
> ---
>
> **Q8**. In section C.1, it is stated that the feature transferability to downstream classification is evaluated without using the projector. Does this mean that the features used are not non-negative?
>
> **A8**. Please see **A2**.
>
> ---
>
> **Q9**. In Appendix C.3, the NCE loss does not include positive samples as it does in the main text. Is this intentional?
>
> **A9**. Sorry, this is a typo. We have fixed it in the revision.
>
> ---
>
> Thank you again for your encouraging comments and valuable feedbacks. We have updated the paper following your suggestions, and we are keen to know whether you find it satisfactory. We are very happy to address your remaining concerns on our work.

---

> ### Comment · Reviewer_eLwm · 2023-11-22
>
> I am grateful to the authors for providing good responses to most of the points I raised, as well as clarifying my misunderstanding of the feature selection experiment in their response A4. Despite this, I still have issues with two of the experiments, and a minor point, that prevent me from changing my recommendation to an accept. I explain these below. Other than these, I feel that the majority of the paper content is worthy of acceptance. My overall recommendation still leads marginally towards reject in its current state (unless there is a convincing response to the following), but I would not complain if the consensus were to accept.
>
> Experiment 4.2 SPARSITY → FEATURE SELECTION
>
> It is not clear that this task is something that would be done in practice: is performance better when subsetting 512 features down to 256, compared to training the model with 256 features originally? Or is there a practical reason for subsetting rather than training a smaller model?
>
> It is clear why NCL would lead each sample to have a more sparse representation, on average, but not why this would lead to the model being more reliant on fewer features, overall (or, at least, this is not clear to me from the paper). Even if it does so empirically, it’s not clear why we should expect this.
>
> Considering both these points, this experiment does not strongly support the purpose of the paper.
>
> 4.3 IN-DOMAIN AND OUT-OF-DOMAIN DOWNSTREAM CLASSIFICATION
>
> The performance when using the projector outputs is not significantly better for NCL than CL. It is only significantly better when using the encoder outputs. As the encoder outputs perform significantly better than the projector outputs, and the projector outputs are the features argued to be disentangled/interpretable, downstream prediction interpretability (for which you would need to use the projector outputs) comes at the cost of performance (for which you would be better off using the encoder outputs). As such, the experiment does not show that NCL leads to a gain in interpretability without a cost in performance, and so does not strongly support the claims of the paper.
>
> Minor point:
>
> Q3/A3
>
> Whilst I appreciate the authors response, the response is more theoretical than I expected. I also appreciate that the latent classes may not be knowable. However, I am still curious: without considering whether it corresponds to a latent class, how many features are there which have at least one sample where (approximately) only that one feature is activated? If this does not seem to happen empirically, it would make for a nice discussion point of the limitations of the theoretical results (in this case: Theorem 5) in practical applications.

---

> > ### Author Response · Authors · 2023-11-22
> > **Further Response to Reviewer eLwm (2/2)**
> >
> > **Q11.** 4.3 IN-DOMAIN AND OUT-OF-DOMAIN DOWNSTREAM CLASSIFICATION
> >
> > > The performance when using the projector outputs is not significantly better for NCL than CL. It is only significantly better when using the encoder outputs. As the encoder outputs perform significantly better than the projector outputs, and the projector outputs are the features argued to be disentangled/interpretable, downstream prediction interpretability (for which you would need to use the projector outputs) comes at the cost of performance (for which you would be better off using the encoder outputs). As such, the experiment does not show that NCL leads to a gain in interpretability without a cost in performance, and so does not strongly support the claims of the paper.
> > >
> >
> > **A11**. When we claim that NCL leads to a gain in interpretability without a cost in performance, our baseline is the original CL’s projection output. **This is true because NCL’s projection output indeed attains similar performance to CL’s, and it also attains better interpretability than NCL as non-negative features.**
> >
> > When comparing internally between NCL’s encoder output and projector output, the encoder output does achieve better downstream accuracy. However, we note that this is a very common phenomenon that appears in almost all CL methods (**not a particular property of NCL** & NCL inherits it from SimCLR). There are some other theories trying to explain this mysterious phenomena [1,2,3], though only limited understanding is obtained under toy settings. Thus, we believe that a full understanding of this phenomenon would require a deep understanding of the role of projection head, while in this work, we mainly focus on the unique properties of NCL in this paper (e.g., the interpretability of projection output).
> >
> > As a last note, even if the encoder output of NCL does not have interpretability, training with NCL objective still improves its downstream performance, showing that learning with a disentangled learning objective also benefits the generalization of intermediate features.
> >
> > **References:**
> >
> > [1] Li et al. Understanding Dimensional Collapse in Contrastive Self-supervised Learning. ICLR. 2022.
> >
> > [2] Gupta et al. Understanding and improving the role of projection head in self-supervised learning. arxiv 2022.
> >
> > [3] Bülent Sarıyıldız et al. No Reason for No Supervision: Improved Generalization in Supervised Models. ICLR 2023.
> >
> > ---
> >
> > **Q12**. I am still curious: without considering whether it corresponds to a latent class, how many features are there which have at least one sample where (approximately) only that one feature is activated?
> >
> > **A12**. To investigate this question, we plot a histogram of the number of activated features of each sample in **Figure 11 (Appendix F.9)**. We can notice that samples generally have much fewer activated dimensions under NCL: **around 70% examples have <20 activated dimensions, and in terms of the smallest, 16 examples are only activated on 2 dimensions.** Considering that the noisy training in practice can lead to some redundant features and some gaps to the optimal solution, we believe that our assumption is indeed approximately true.
> >
> > ---
> >
> > Thanks for your insightful questions. We have carefully addressed your concerns above with extended experiment results (now included in the paper), which makes the work more complete. We respectfully wish that you could re-evaluate our work based on the updated results.  Please let us know if there is more to clarify.

---

> > > ### Comment · Reviewer_eLwm · 2023-11-22
> > >
> > > I would like to thank the authors for their responses. In light of their responses and the updates to the manuscript, I have decided to increase my score to 6. The paper could be more convincing in its experiments, but the overall contribution leads the paper to be marginally above the acceptance threshold.

---

> ### Author Response · Authors · 2023-11-22
> **Further Response to Reviewer eLwm (1/2)**
>
> Thanks for your quick response and for your insightful questions! We are happy to address your remaining concerns.
>
> ---
>
> **Q10**. Experiment 4.2 SPARSITY → FEATURE SELECTION
>
> **A10.** We will address your specific questions point by point.
>
> > It is not clear that this task is something that would be done in practice: is performance better when subsetting 512 features down to 256, compared to training the model with 256 features originally? Or is there a practical reason for subsetting rather than training a smaller model?
> >
>
> **A real-world scenario for feature selection.** We consider the example of real-time image/text retrieval, which is a common application scenario in search engine and recommendation system. Upon real-time request burden and computation budget, we may need to adjust the retrieval cost adaptively to reduce the latency. A viable approach is through selecting a few feature dimensions to reduce the cost, and the number of selected features may **change dynamically to tradeoff between accuracy and latency**.
>
> **Gain of NCL.** For such scenarios, it would be costly to learn multiple models of different output dimensions, and NCL with disentangled features can come to an aid. To show this, we conduct a new experiment (**Appendix F.8**) to examine the performance of image retrieval across different output dimensions (512→256). From **Figure 10**, we can see that TA-based NCL features keep a high level of mAP and **have almost no degradation of performance**. Instead, **CL features keep degradation under fewer features and underperform NCL a lot at last.** This experiment shows the clear advantage of the sparsity and disentanglement property of NCL.
>
> **Compared to separate model**. Besides, we can find that NCL also performs much better when we subset 256 features from a 512-d model (11.2 mAP), rather than using a 256-d model from scratch (9.4 mAP). Thus, the feature selection method is helpful for both dynamic-dimension retrieval and final performance under the same dimension.
>
> > It is clear why NCL would lead each sample to have a more sparse representation, on average, but not why this would lead to the model being more reliant on fewer features, overall (or, at least, this is not clear to me from the paper). Even if it does so empirically, it’s not clear why we should expect this.
> >
>
> We are sorry for the potential confusion. As illustrated in Figure 3, the sparse representations of NCL are only sparsely activated on a few dimensions as well. Upon a close examination, we find that many feature dimensions are only activated with a few examples, even none (86 of 512 dimensions are never activated). It means that these features are far less common and can be discarded without much influence.
>
> **Reasons.** Theoretically, we have shown that to attain the optimal solution (Eq. 8), we only need the feature dimension to be the number of all latent classes, and other dimensions beyond that would be unnecessary and filled with zero (footnote 3). This explains why practical NCL algorithms learn some (approximately) null features. Based on this property, we can select a subset of dimensions from NCL features while largely keeping its original performance. We have added elaboration on this part in **Sec 4.2**.

---

### Official Review · Reviewer_R2Pb · 2023-11-01

**Soundness:** 3 good
**Presentation:** 3 good
**Contribution:** 2 fair
**Rating:** 6
**Confidence:** 3

**Summary:**

The manuscript introduces a new self-spervised paradigm known as non-negative contrastive learning (NCL). Drawing inspiration from spectral contrastive learning (SCL) by HaoChen et al., 2021 and non-negative matrix factorization (NMF), the authors proposed a contrastive loss that enforces non-negativity constraints on the extracted features.

The authors demonstrate that, assuming minimal class overlap, NCL has the potential to learn a set of sparse features with low correlation. Under a one-hot representation, NCL achieves orthogonality among its features (Theorem 4). Furthermore, when at least one sample is uniquely assigned to each class, NCL learns a set of distinguishable and disentangled features (Theorem 5).

To validate the effectiveness of NCL, the authors conducted experiments on three benchmark datasets: CIFAR-10, CIFAR-100, and ImageNet-100, comparing it with a conventional contrastive learning methods, i.e. SimCLR. Across all datasets, NCL's features exhibit greater sparsity, consistency, and less entanglement, leading to improved representation of class identity in classification tasks.

**Strengths:**

- The manuscript is generally well written.
- The proposed contrastive learning paradigm is theoretically grounded, with a connection to NMF problem.

**Weaknesses:**

- **Novelty:** While the author has made a great effort to establish a connection betweenNMF and contrastive learning in the NCL approach, it appears that NCL can be seen as an extension of SCL with non-negativity constraints and similar justifications and proofs.

- **Contribution:** The reported results in Table 2, and Figure 5 do not convincingly support a significant improvement brought about by NCL compared to the classical contrastive learning method. Furthermore, the absence of a direct comparison between NCL and SCL is notable.

- **Generality and applicability of NCL's representation:** While the authors have theoretically demonstrated that NCL can yield sparse and less entangled representations suitable for downstream tasks like clustering and classification, the critique is that the suggested definition for an optimal and interpretable representations does not necessarily covers the broader goal of capturing all latent factors within the data structure. In many practical applications, e.g. in VAE studies or mixture modeling, the aim is to obtain comprehensive representations that also cover continuous and non-sparse variabilities, which remains unaddressed in this work.

I appreciate the problem addressed by the author and introducing the constrained contrastive learning, which I find intriguing. The manuscript seems to have established a reasonable foundation. However, I think there are certain concerns that need to be addressed. I am open to revising my evaluation pending the author's response.

**Questions:**

- It seems there might be a typo error in the proof of Theorem 2. It appears that the negative sample $x^-$ should be the positive one, $x^+." right?

- If the primary objective of introducing non-negativity constraints is to encourage sparser representations, have you considered whether regularizing SCL loss with a sparsity regularizer might produce similar results?

- Could you clarify whether the results presented in Tables 1 and 2, as well as Figure 5, are based on a single round of experimentation or multiple iterations? When the text mentions *"on average"* it would be helpful to specify the number of runs that the average is calculated over.

- In section 4.1, Results, it is mentioned that *"The advantage is larger by considering the top features... since learned features also contain noisy dimensions."* This statement appears to be somewhat contradictory to the claim of a less-entangled representation.

---

> ### Author Response · Authors · 2023-11-19
> **Response to Reviewer R2Pb (1/2)**
>
> Thanks for your detailed reading and appreciating the clarity and theory of our work. Below, we will elaborate on the difference to previous works and address your concerns in detail.
>
> ---
>
> **Q1**. **Novelty:** It appears that NCL can be seen as an extension of SCL with non-negativity constraints and similar justifications and proofs.
>
> **A1**. We are afraid that we do not think that our analysis uses “similar justifications and proofs” to SCL. Indeed, NCL is a variant of SCL by incorporating the non-negative constraint. However, we highlight that **adding this constraint makes this problem have very different structures (optimal solutions,  theoretical properties, empirical behaviors), and thus requires very different techniques to deal with**.
>
> **Differences.** Note that the extension from MF to NMF is highly nontrivial. The two problems have very different properties and require different algorithms, leading to a new branch of research area. Because of the equivalence between MF/NMF and CL/NCL, our extension from CL to NCL is also technically non-trivial and has practical significance. To be more specific,
>
> - **Different optimal solutions.** CL optimal features are **eigenvectors** of the augmentation graph [1], which usually contain negative values that are infeasible solutions for NCL; in turn, it also means that NCL optimal features enforcing the non-negativity constraint might be non-optimal for CL. Thus, **the old techniques for analyzing CL through eigendecomposition [1] cannot be applied to NCL.**
> - **Different theoretical properties.** CL optimal features are eigenvectors that are non-unique (no identifiability), dense (no sparsity), and rotationally symmetric (no dimensional semantic consistency). In comparison, NCL optimal features enjoy these properties, showing their clear difference.
> - **Different empirical properties.** As also verified in practice, CL features do not preserve these interpretability-related properties as NCL features, including semantic consistency, sparsity, and feature disentanglement.
>
> **Solutions.** To account for their differences, in Sec 3, we **have developed a new latent-class-based theoretical framework** to derive and analyze the optimal solutions of NCL, and guarantee its downstream performance and interpretability. This is not possible within the original analysis framework of Haochen et al [1] or Arora et al [2] alone. Instead, it requires us to bridge two frameworks and characterize optimal solutions of the constrained problem under this new framework, which may be of independent interest.
>
> Overall, we believe that our theoretical analysis is also technically novel, since it builds a new framework for analyzing non-negative constrained CL methods for the first time. Although simple, the non-negative constraint extension could be deep and meaningful to the field.
>
> **References:**
>
> [1] Haochen et al. Provable Guarantees for Self-Supervised Deep Learning with Spectral Contrastive Loss. In NeurIPS. 2021.
>
> [2] Arora et al. A Theoretical Analysis of Contrastive Unsupervised Representation Learning. In ICML. 2019.
>
> ---
>
> **Q2**. **Contribution:** The reported results in Table 2, and Figure 5 do not convincingly support a significant improvement brought about by NCL compared to the classical contrastive learning method. Furthermore, the absence of a direct comparison between NCL and SCL is notable.
>
> **A2**. **Main Contribution.** As we emphasized extensively in the paper (esp. Abstract and Introduction), **the** **main focus of this paper is NOT to improve downstream performance, but to resolve the feature interpretability problem** of contrastive learning. Interpretability is an important aspect of representation learning, as acknowledged by Reviewer kpGZ, *“Interpretability of ML methods and learned representations is an important problem.”*.
>
> **Comparison.** Therefore, the comparable performance of Table 2 does not conflict with the effectiveness of NCL on improving feature interpretability. Instead, since many interpretability methods often come at the cost of not-so-good representation quality (such as, disentangled/identifiable VAE methods), the fact that NCL can maintain the same level of performance as canonical CL while attaining interepretability makes it **a simple and desirable solution to attain interpretability with no sacrifice on performance**.
>
> To this end, we respectfully suggest that you could evaluate our work based on its contribution on enhancing feature interpretability in representation learning.
>
> **Comparison between SCL and NCL**. Following your suggestion, we further compare SCL and NCL in terms both feature interpretability, and results are included in **Appendix F.1**. As quoted below, NCL can also attain slight improvement over SCL, while bringing clear interpretability improvements in terms of class consistency, sparsity, and correlation. It shows that NCL indeed works well across different CL objectives.
>
> (continue below)

---

> ### Author Response · Authors · 2023-11-19
> **Response to Reviewer R2Pb (2/2)**
>
> (continue A2 above)
>
> *Comparison of SCL and its NCL version on CIFAR-10.*
>
> | Objective | Linear Accuracy | Semantic Consistency | Sparsity | Correlation Error |
> | --- | --- | --- | --- | --- |
> | SCL | 88.0 | 19.0 | 0.002 | 0.117 |
> | NCL | 88.6 | 24.0 | 64.654 | 0.004 |
>
> **Reference:**
>
> [1] Zhang et al. Improving VAE-based Representation Learning. arxiv 2022.
>
>
> ---
>
> **Q3**. **Generality and applicability of NCL's representation:** While the authors have theoretically demonstrated that NCL can yield sparse and less entangled representations suitable for downstream tasks like clustering and classification, the critique is that the suggested definition for an optimal and interpretable representations does not necessarily covers the broader goal of capturing all latent factors within the data structure. In many practical applications, e.g. in VAE studies or mixture modeling, the aim is to obtain comprehensive representations that also cover continuous and non-sparse variabilities, which remains unaddressed in this work.
>
> **A3**. Good question! In this paper, following the convention of CL theory, we also consider linear classification as the downstream task and thus consider sparse features. Nevertheless, **our theoretical framework in Sec 3 can be easily extended to continuous and non-sparse features as well as mixture modeling**. In fact, the latent variable $c$ is generally defined and can also take continuous values. As long as the data generation process holds (Assumption 1) and $c$ is non-negative, Theorem 3 still gives a characterization of the optimal solutions, showing that NCL can recover the ground-truth posterior distribution of latent classes $c$.
>
> Thus, NCL is not limited to learning discrete classes, but also applies to continuous settings as long as the latents are non-negative. We have added a footnote on **page 6** to discuss this part.
>
> ---
>
> **Q4**. It seems there might be a typo error in the proof of Theorem 2. It appears that the negative sample.
>
> **A4**. Thanks! We have fixed it.
>
> ---
>
> **Q5**. If the primary objective of introducing non-negativity constraints is to encourage sparser representations, have you considered whether regularizing SCL loss with a sparsity regularizer might produce similar results?
>
> **A5**. Indeed, it is an interesting idea to encourage sparser representations with a sparsity regularizer. To compare it with NCL, we conduct additional experiments by adding the l1 norm to the original contrastive objective. As shown in the following table, by simply adding a sparsity regularizer, the feature sparsity (calculated as in Figure 5(b)) is much lower than NCL. Meanwhile, the downstream performance even performs worse than original contrastive learning. We have included this discussion in **Appendix F.2**.
>
> *Comparing NCL with $\ell_1$ regularization on CIFAR-10.*
>
> | Objective | Linear Accuracy | Feature Sparsity |
> | --- | --- | --- |
> | SCL | 88.0 | 0.002 |
> | NCL | 88.6 | 69.654 |
> | SCL+l1 norm | 87.2 | 0.031 |
>
> ---
>
> **Q6**. Could you clarify whether the results presented in Tables 1 and 2, as well as Figure 5, are based on a single round of experimentation or multiple iterations? When the text mentions *"on average"* it would be helpful to specify the number of runs that the average is calculated over.
>
> **A6**. To be more rigorous, we have replaced all benchmark results with mean and stdev values obtained from multiple runs (see updates in Tables 1-4). Due the limit of time, we run 3 random trials. It can be seen that NCL indeed often outperforms CL significantly on feature selection and downstream tasks.
>
> In Sec 4.3, “on average” means that the score is taken over three different datasets (as an overall improvement) in Table 3, not over random trials. We have added explanations in the revision.
>
> ---
>
> **Q7**. In section 4.1, Results, it is mentioned that *"The advantage is larger by considering the top features... since learned features also contain noisy dimensions."* This statement appears to be somewhat contradictory to the claim of a less-entangled representation.
>
> **A7**. Here, by “noisy”, we are referring to those class-irrelevant features with small eigenvalues, which frequently appear in most representation learning algorithms. These noisy dimensions have neglectable influence on the final performance, and it might be introduced by the insufficient stochastic learning in practice. To eliminate their influence, **it is a common practice to consider only the top features when evaluating identifiability, as also adopted in [1].**
>
> [1] Roeder et al. On linear identifiability of learned representations.  In ICML. 2021.
>
> ---
>
> Thank you again for your careful reading. We have carefully refined our paper according to your suggestions, and address each of your concerns above. We respectfully suggest that you could re-evaluate our work based on these updated results. We are very happy to address your remaining concerns on our work.

---

> ### Author Response · Authors · 2023-11-22
>
> Dear Reviewer R2Pb,
>
> We have carefully prepared a detailed response to address each of your questions. Would you please take a look and let us know whether you find it satisfactory?
>
> We note that Reviewer kpGZ has appreciated our response and raised the score beyond the acceptance bar. We also respectfully suggest that you could re-evaluate our work with the updated explanations and results.
>
> Thanks! Have a great day!
>
> Authors

---

> ### Comment · Reviewer_R2Pb · 2023-11-22
>
> I appreciate the author's efforts in presenting a detailed rebuttal alongside additional experiments, which addressed most of my questions.  Considering the effort in the rebuttal, I increased my score.
>
> Moreover, I suggest that the authors integrate the SCL and SCL+L1 outcomes into the main body of the text for better coherence.
>
> However, following the authors' response, I still have a few remaining questions.
> - The authors mentioned *"the primary aim of this paper is not to enhance downstream performance but to address the interpretability issue regarding features"*. Here, I am uncertain about the precise meaning of **"interpretability"**. Is it about achieving a more disentangled and sparse features? In the context of the classification task, a latent space with highly disentangled factors, representing distinct class identities, would theoretically yield significantly higher accuracy compared to CL and SCL lacking such disentanglement or sparsity. How do we justify interpretability in this context? What benefits or insights do we derive from this particular definition of "interpretability"?
>
> - Revisiting my concerns regarding *"Generality"*, I am still uncertain about how the current formulation, under Assumption 1, can be extended to a continuous representation. Assumption 1 implies conditional independence among samples in a class. How does this assumption hold when **c** is not a discrete class label?

---

> ### Author Response · Authors · 2023-11-23
> **Further Response to Reviewer R2Pb**
>
> Thanks for appreciating our response and for raising the score! We will address your remaining concerns below, and hope you find it satisfactory.
>
> ---
>
> **Q8**. Moreover, I suggest that the authors integrate the SCL and SCL+L1 outcomes into the main body of the text for better coherence.
>
> **A8**. Indeed, the comparison between NCL and $\ell_1$ regularization clearly demonstrates the benefits of NCL. We will merge it into the main paper with some rearrangements of space.
>
> ---
>
> **Q9**. The authors mentioned *"the primary aim of this paper is not to enhance downstream performance but to address the interpretability issue regarding features"*. Here, I am uncertain about the precise meaning of **"interpretability"**. Is it about achieving a more disentangled and sparse features?
>
> **A9**. As you suggested, formally, in this paper, we use “interpretability” to refer to a more *disentangled and sparse* features. Further, intuitively, we use this word to highlight that NCL features are more *human inspectable*, as shown in Figure 1.
>
> > In the context of the classification task, a latent space with highly disentangled factors, representing distinct class identities, would theoretically yield significantly higher accuracy compared to CL and SCL lacking such disentanglement or sparsity. How do we justify interpretability in this context? What benefits or insights do we derive from this particular definition of "interpretability"?
> >
>
> In fact, we would note that *disentanglement itself would not necessarily improve linear classification accuracy*, since the linear head can absorb the rotational symmetry.  To see this, consider a disentangled and sparse feature $z$, and a rotated one as $z’=Rz$ (now entangled and non-sparse), where $R$ is a rotation matrix. If the disentangled feature $z$ permits an optimal linear classifier $g(z)=Bz$, the rotated feature $z’$ would also permit a classifier $g’(z’)=(BR^\top) z’$, which has exactly the same prediction and accuracy as $g(z)$. Based on this observation, we believe that there is no necessary connection between improving disentanglement and improving accuracy. So, we only expect CL and NCL to attain comparable linear accuracy (Sec 4.3).
>
> Meanwhile, **improving disentanglement would certainly benefit the scenarios that does require feature interpretability, especially those involving humans in the loop**. As shown in Figure 1, NCL features are more human inspectable, making them more trustworthy, editable, and accountable. For example, NCL features are easier for selecting useful/reliable features with human priors. Leveraging the disentanglement property, we also show in Sec 4.2 that NCL has larger advantages under automatic feature selection.
>
> ---
>
> **Q10**. Revisiting my concerns regarding *"Generality"*, I am still uncertain about how the current formulation, under Assumption 1, can be extended to a continuous representation. Assumption 1 implies conditional independence among samples in a class. How does this assumption hold when **c** is not a discrete class label?
>
> **A10**. Sorry for the confusion. Let us further explain it. When $c$ denotes a non-negative continuous variable, such as, the latent variable as in VAE, we can assume access to two observed samples $x,x^+$ drawn independently from the same latent $c$. In such case, the formulation in Assumption 1 also holds, since  $\forall x, x^{\prime} \in \mathcal{X}, \mathcal{P}\left(x, x^{\prime}\right)=\mathbb{E}_c \mathcal{P}(x \mid c) \mathcal{P}\left(x^{\prime} \mid c\right)$, where $P(x|c)$ denote the probabilistic decoder. So, our theoretical framework still applies to the continuous setting.
>
> Meanwhile, thanks to your reminder, we do notice that the optimal representation for the continuous problem may differ from that of discrete problems in Eq.8. An approximate solution is to quantize $c$ into finite discrete values (as done in Haochen et al.), then Eq. 8 still applies. In this paper, we mainly focus on the discrete and finite case with analogy to the canonical matrix factorization scenario. We also notice that there are some *continuous* generalization of MF/NMF [1,2], which would be a viable approach to directly characterize the continuous latent variable in NCL. We will discuss more on it in the revision.
>
> **References**:
>
> [1] Townsend, Alex, and Lloyd N. Trefethen. "Continuous analogues of matrix factorizations." *Proceedings of the Royal Society A: Mathematical, Physical and Engineering Sciences* 471.2173 (2015): 20140585.
>
> [2] Hautecoeur, Cécile, and François Glineur. "Nonnegative matrix factorization over continuous signals using parametrizable functions." *Neurocomputing* 416 (2020): 256-265.
>
> ---
>
> Thanks for your careful reading and for appreciating our response. Hope the clarification above could address your concerns. Please let us know if there is more to clarify.

---

### Official Review · Reviewer_Szt3 · 2023-11-01

**Soundness:** 2 fair
**Presentation:** 2 fair
**Contribution:** 2 fair
**Rating:** 5
**Confidence:** 4

**Summary:**

The paper propose Non-negative Contrastive Learning (NCL), which is inspired by Non-negative Matrix Factorization (NMF), to derive interpretable features. Specifically, NCL add a non-negative transformation at the end of a standard encoder to generate non-negative features. The paper also demonstrates the advantages of NCL by experiments.

**Strengths:**

The writing of this paper is clear, and the descriptions and justifications of the methods are comprehensible.

**Weaknesses:**

1. When presenting key arguments in the paper, some claims lack proper foundation, while others are based solely on partial visualization results from a specific dataset. I remain somewhat skeptical of these points.
2. The results in Table 2 of the article indicate that the improvement from the proposed method is rather limited. I remain uncertain about the method's true efficacy.

**Questions:**

1. In the first paragraph of sec 1, the authors show top activated examples along each feature dimension to demonstrate the interpretability of the features. Is this a common practice? Why is this approach considered reasonable?
2. In the second paragraph of sec 2.2, why does the rotational symmetry in the optimal solution hurt the performance?
3. The proposed method, simply adding a relu layer after the encoder, seems to be too easy to improve the performance.

---

> ### Author Response · Authors · 2023-11-19
> **Response to Reviewer Szt3 (1/2)**
>
> Thanks for your review, but we are afraid that there are potentially some misunderstandings on the main contributions of this work, which is on **feature** **interpretability** instead of performance. Below, we give a detailed elaboration on this point, and address your concerns point by point.
>
> ---
>
> **Q1.** When presenting key arguments in the paper, some claims lack proper foundation, while others are based solely on partial visualization results from a specific dataset. I remain somewhat skeptical of these points.
>
> **A1**. It appears to us that the concerns here correspond to the first two questions in the Question section (**Q3** about visualization and **Q4** about rotational symmetry). We will answer them in detail below. Please do let us know if there are other specific parts of the theoretical or empirical results that you still find skeptical after reading our response.
>
> ---
>
> **Q2**. The results in Table 2 of the article indicate that the improvement from the proposed method is rather limited. I remain uncertain about the method's true efficacy.
>
> **A2**. As we emphasized extensively in the paper (esp. Abstract and Introduction), **the** **main focus of this paper is NOT to improve downstream performance, but to resolve the feature interpretability problem** of contrastive learning. Interpretability is an important aspect of representation learning, as acknowledged by Reviewer kpGZ, *“Interpretability of ML methods and learned representations is an important problem.”*.
>
> Therefore, the comparable performance of Table 2 does not conflict with the effectiveness of NCL on improving feature interpretability. Instead, since many interpretability methods often come at the cost of not-so-good representation quality (for example, VAEs only get ~70% linear probing accuracy on CIFAR-10 [1]), the fact that NCL can maintain the same level of performance as canonical CL (87.8% linear accuracy) while attaining feature interpretability makes it **a simple and desirable solution to attain interpretability with no sacrifice on performance**.
>
> To this end, we respectfully suggest that you could evaluate our work based on its contribution on enhancing feature interpretability in representation learning.
>
> **Reference:**
>
> [1] Zhang et al. Improving VAE-based Representation Learning. arxiv 2022.
>
> ---
>
> **Q3**. In the first paragraph of sec 1, the authors show top activated examples along each feature dimension to demonstrate the interpretability of the features. Is this a common practice? Why is this approach considered reasonable?
>
> **A3**. As discussed in Bengio et al. [1], each feature dimension should be disentangled and has its own meaning. Encoder-decoder methods (like VAEs) usually validate this point by generating samples with varying latent factors. For encoder-only methods (like CL), a viable approach is to **examine whether training samples activated in each dimension have similar semantics**. In our paper, we have used two approaches to examine this:
>
> - **Qualitative visualization**. Since samples with top activations indicate strong signals of this feature, we select 10 top activated samples for a clear demonstration in **Figure 1**. If these top activated samples even have distinct semantics (like CL), it means a lack of feature interpretability. We notice that a recent NeurIPS paper [2] ***also uses this approach to inspect the semantics of each feature*** (see Figure 1 in [2]). We have further added **CIFAR-100 and ImageNet-100 examples in Appendix F.6**, where NCL also has clear advantages over CL on feature interpretability.
> - **Quantitative measure**. To quantitatively measure this interpretability property, we also design a metric called **“Semantic Consistency”,** which calculates the proportion of all activated samples that belong to their most frequent class along each feature dimension (see Fig 4a caption). As shown in Fig 4a, NCL also outperforms CL by a large margin on semantic consistency. Besides, we have also **quantitatively evaluated feature sparsity, correlation, and disentanglement** for a comprehensive analysis (Sec 3 & 4). These measures all show that NCL has much better interpretability than CL.
>
> **Reference:**
>
> [1] Bengio et al. Representation learning: A review and new perspectives. IEEE Transactions on Pattern Analysis and Machine Intelligence, 35(8):1798–1828, 2013.
>
> [2] Zhang et al. Identifiable Contrastive Learning with Automatic Feature Importance Discovery. In NeurIPS. 2023.

---

> ### Author Response · Authors · 2023-11-19
> **Response to Reviewer Szt3 (2/2)**
>
> **Q4**. In the second paragraph of sec 2.2, why does the rotational symmetry in the optimal solution hurt the performance?
>
> **A4**. We are afraid that there are some misunderstandings here. **The problem of rotational symmetry is that it causes the loss of CL’s feature interpretability, not downstream performance.** Theoretically, rotational symmetry has no influence on downstream performance since these solutions are equivalently optimal [1]. However, this property also makes CL features lack semantic consistency, sparsity, and disentanglement (elaborated in Sec 2.2). By introducing the non-negative constraint, we show that **NCL can restore these interpretability properties while retaining downstream performance, both theoretically and empirically.**
>
> **References:**
> [1] Haochen et al. Provable Guarantees for Self-Supervised Deep Learning with Spectral Contrastive Loss. In NeurIPS. 2021.
>
> ---
>
> **Q5**. The proposed method, simply adding a relu layer after the encoder, seems to be too easy to improve the performance.
>
> **A5**. As we clarified in **A2**, the main goal of this work is **to improve feature interpretability instead of downstream performance**, and we have shown that NCL has accomplished this goal with **extensive theoretical and empirical justifications in Sec 3 & 4**.
>
> We believe that **in terms of achieving the same goal, a simple, general, and easy-to-use solution would be a pro rather than a con, according to Occam's Razor**. We would humbly quote from a blog by Michel J. Black [1] on his view of simple methods:
>
> > The simplicity of an idea is often confused with a lack of novelty when exactly the opposite is often true.  A common review critique is
> >
> >
> > > *The idea is very simple. It just changes one term in the loss and everything else is the same as prior work.*
> > >
> >
> > If nobody thought to change that one term, then it is *ipso facto* novel. The inventive insight is to realize that a small change could have a big effect and to formulate the new loss.
> >
>
> With the ReLU reparameterization, we introduce minimal and model-agnostic change to existing CL methods, so it can be easily incorporated to any existing frameworks to greatly enhance feature interpretability, while maintaining the same level of downstream performance. As noted by Reviewer kpGZ, it is *“a cool idea and innovative”.* Reviewer eLwm commented that *“Although the derivation are in-depth, the proposed modification to CL is fairly simple to understand and implement.”*
>
> Thus, we believe that the simplicity of NCL would be an advantage to **attain much better feature interpretability while maintaining the same level of performance**.
>
> **Reference:**
>
> [1] Michael J. Black. Novelty in Science. https://perceiving-systems.blog/en/post/novelty-in-science.
>
> ---
>
> Thank you again for your review. We have carefully refined our paper according to your suggestions, and address each of your concerns above. We respectfully suggest that you could re-evaluate our work based on these updated results. We are very happy to address your remaining concerns on our work during the discussion stage.

---

> ### Author Response · Authors · 2023-11-22
>
> Dear Reviewer Szt3,
>
> We have carefully prepared a detailed response to address each of your questions. Would you please take a look and let us know whether you find it satisfactory?
>
> We note that Reviewer kpGZ has appreciated our response and raised the score beyond the acceptance bar. We also respectfully suggest that you could re-evaluate our work with the updated explanations and results.
>
> Thanks! Have a great day!
>
> Authors

---

### Author Response · Authors · 2023-11-19
**Paper Update**

We sincerely thank all reviewers for their detailed reading and valuable comments. We have carefully responded their concerns, and incorporated these suggestions in the updated manuscript with 3 extended pages. The main revisions are:

- Sec 2: revise preliminary to be more rigorous and readable.
- Sec 3.2: add discussion on applicability to continuous variables.
- **Sec 4: report mean and stdev results obtained from random runs for all experiment results (Table 1-4)**
- Appendix C.3: update InfoNCE formulation to be consistent with the original one
- **Appendix F (new!)**: **add many new analysis results:**
    - 1) performance on spectral contrastive learning,
    - 2) comparison to $\ell_1$ sparse regularization,
    - 3) disentanglement on synthetic data,
    - 4) examination of optimal features,
    - 5) performance of projector output,
    - 6) more visualization samples from CIFAR-100 and ImageNet-100.
    - 7) verification of theoretical assumptions
    - 8) performance under dynamic output lengths

We note that we change Figure 2 to Tabe 2 and split the original Table 2 into two for better clarity. So the numbering of figures and tables after that is affected. For example, Table 2 now becomes Tables 3 & 4.

---

### Meta-Review · Area_Chair_Cb5y · 2023-12-07

**Metareview:**

Thanks for your submission to ICLR.

This paper considers a non-negative matrix factorization (NMF) type approach applied to contrastive learning, building on success of NMF and its use in terms of interpretability and theoretical power.

Initially, the reviewers were mixed, leaning possibly reject on this paper.  Some of the initial concerns included that some claims lacked proper foundation, there were some limited/missing results, and there was a concern that the overall contribution was somewhat limited.  The rebuttal helped significantly, with multiple reviewers raising their scores.  Ultimately, three of the four reviewers were willing to advocate for accepting this paper.  The reviewer who still had a score of 5 never responded to the rebuttal but the authors did address the concerns.

Overall I am now willing to recommend accepting the paper.

**Justification For Why Not Higher Score:**

It's an interesting paper but the scores are fairly borderline; I would not recommend pushing up to spotlight.  Further, there are some lingering concerns of the first reviewer that may still need to be addressed.

**Justification For Why Not Lower Score:**

The authors did a very nice job responding to the criticisms of the reviewers, and two of the reviewers raised their score.  The only reviewer who did not give an accept score did not actually participate in the discussion.

---

### Decision · Program_Chairs · 2024-01-16

Accept (poster)